biomathematics/mathematical modelling

COVID-19, reproduction number

**Author for correspondence:**
Tomasz Lipniacki
e-mail: tlipnia@ippt.pan.pl

# Super-spreading events initiated the exponential growth phase of COVID-19 with $\mathcal{R}_0$ higher than initially estimated

Marek Kochańczyk[1], Frederic Grabowski[2]
and Tomasz Lipniacki[1]

[1]Department of Biosystems and Soft Matter, Institute of Fundamental Technological Research, Polish Academy of Sciences, 02-106 Warsaw, Poland
[2]Faculty of Mathematics, Informatics and Mechanics, University of Warsaw, 02-097 Warsaw, Poland

MK, 0000-0003-1215-3920; FG, 0000-0003-4070-9500;
TL, 0000-0002-3488-2561

The basic reproduction number $\mathcal{R}_0$ of the coronavirus disease 2019 has been estimated to range between 2 and 4. Here, we used an SEIR model that properly accounts for the distribution of the latent period and, based on empirical estimates of the doubling time in the near-exponential phases of epidemic progression in China, Italy, Spain, France, UK, Germany, Switzerland and New York State, we estimated that $\mathcal{R}_0$ lies in the range 4.7–11.4. We explained this discrepancy by performing stochastic simulations of model dynamics in a population with a small proportion of super-spreaders. The simulations revealed two-phase dynamics, in which an initial phase of relatively slow epidemic progression diverts to a faster phase upon appearance of infectious super-spreaders. Early estimates obtained for this initial phase may suggest lower $\mathcal{R}_0$.

## 1. Introduction

The basic reproduction number $\mathcal{R}_0$ is a critical parameter characterizing the dynamics of an outbreak of an infectious disease. By definition, $\mathcal{R}_0$ quantifies the expected number of secondary cases generated by an infectious individual in an entirely susceptible population. $\mathcal{R}_0$ may be influenced by natural conditions (such as seasonality) as well as socio-economic factors (such as population density or ingrained societal norms and practices) [1]. Accurate estimation of $\mathcal{R}_0$ is of crucial importance

because it informs the extent of control measures that should be implemented to terminate the spread of an epidemic. Also, $\mathcal{R}_0$ determines the immune proportion $f$ of population that is required to achieve herd immunity, $f = 1 - 1/\mathcal{R}_0$.

A preliminary estimate published by the World Health Organization (WHO) suggested that $\mathcal{R}_0$ of coronavirus disease 2019 (COVID-19) lies in between 1.4 and 2.5 [2]. Later this estimate has been revised to 2–2.5 [3], which is broadly in agreement with numerous other studies that, based on official data from China, implied the range of 2–4 (see Liu *et al.* [4] or Boldog *et al.* [5] for a summary). This range suggests an outbreak of a contagious disease that should be containable by imposition of moderate restrictions on social interactions. Unfortunately, moderate restrictions that were implemented in e.g. Italy or Spain turned out to be insufficient to prevent a surge of daily new cases and, consequently, nationwide quarantines had to be introduced.

We estimated the range of $\mathcal{R}_0$ of COVID-19 based on the doubling times observed in the exponential phases of the epidemic in China, Italy, Spain, France, UK, Germany, Switzerland and New York State. For each of these locations, we used trajectories of both cumulative confirmed cases and deaths [6]. Since our stochastic simulations suggested that the epidemic may have two-phase dynamics—slow (and susceptible to extinction) before any super-spreading events occur, and fast and steadily expanding after the occurrence of super-spreading events—to capture the second phase of the trajectories, we analysed them after a fixed threshold of cases or deaths had been exceeded, in two-week intervals. Both the stochastic simulations and $\mathcal{R}_0$ estimates were obtained within a susceptible–exposed–infected–removed (SEIR) model that correctly reproduces the shape of the latent period distribution and yields a plausible mean generation time. We concluded that the range of $\mathcal{R}_0$ is 4.7–11.4, which is considerably higher than most early estimates. We conjecture that these early estimates were obtained for the first phase of the epidemic in which super-spreading events were absent.

## 2. Results

### 2.1. The SEIR model

We used an SEIR model (see Methods for model equations and justification of parameter values) in which:

— we assumed that the latent period is the same as the incubation period and is Erlang-distributed with the shape parameter $m = 6$ and the mean of 5.28 days $= 1/\sigma$ [7];
— we assumed that the infectious period is Erlang-distributed with the shape parameter $n = 1$ (exponentially distributed) or $n = 2$, and the mean of 2.9 days $= 1/\gamma$ [8,9];
— the infection rate coefficient $\beta$ was determined from $\sigma$, $\gamma$, $m$, $n$ and doubling time $\mathcal{T}_d$, which in turn was estimated based on the epidemic data as described in the next subsection, ultimately allowing us to estimate $\mathcal{R}_0 = \beta/\gamma$ as $\mathcal{R}_0(\mathcal{T}_d)$.

The use of the Erlang distributions directly translates to the inclusion of multiple consecutive substates in the SEIR model, meaning that we assumed $m$ 'exposed' substates and $n$ 'infectious' substates (Erlang distribution is a distribution of a sum of independent exponentially distributed variables of the same mean).

### 2.2. Estimation of $\mathcal{R}_0$ in the exponential growth phase

First, we estimated the doubling time $\mathcal{T}_d$ within two-week periods beginning on the day in which the number of confirmed (in the SEIR model naming convention, 'removed', see Methods) cases exceeded 100 or the number of deaths exceeded 10 in China, six European countries and New York State (figure 1*a,b*). Values of $\mathcal{T}_d$ that we obtained lie in between $\mathcal{T}_d^{\min} = 1.86$ days (based on cases in New York State) and $\mathcal{T}_d^{\max} = 2.96$ days (based on deaths in Switzerland).

Then, we estimated the range of $\mathcal{R}_0$ as a function of the doubling time $\mathcal{T}_d$ using a formula that takes into account the mean latent and infectious period, $1/\sigma$ and $1/\gamma$, respectively, as well as the shape parameters $m$ and $n$, see equation (4.8) in Methods. The lower bound has been obtained using the model variant with $n = 2$ (two 'infectious' substates), whereas the upper bound results from the model with $n = 1$ (one 'infectious' substate), figure 1*c*. After plugging $\mathcal{T}_d^{\max}$ and $\mathcal{T}_d^{\min}$ in, respectively, the variant of our model with the lower $\mathcal{R}_0(\mathcal{T}_d)$ curve ($n = 2$) and the variant with the higher $\mathcal{R}_0(\mathcal{T}_d)$ curve ($n = 1$), we arrived at the estimated $\mathcal{R}_0$ range of 4.7–11.4. The cases-based doubling time for China, 2.36, is consistent with the value of 2.4 reported by Sanche *et al.* [11], who estimated that $\mathcal{R}_0$

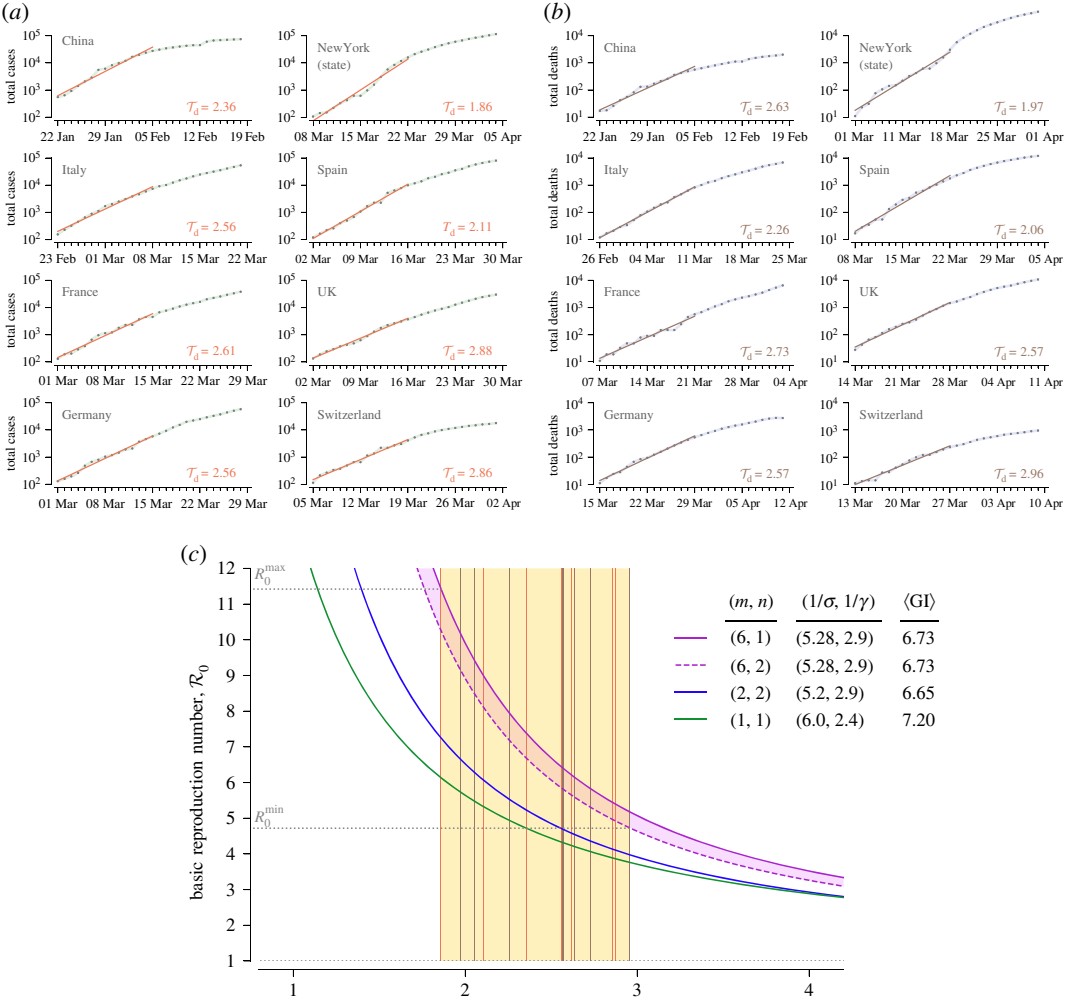

**Figure 1.** Estimation of the doubling time and the resulting basic reproduction number $\mathcal{R}_0$. (a,b) Estimates of the doubling time $\mathcal{T}_d$ for China, six European countries and New York State using two-week periods beginning (a) when the number of confirmed cases exceeds 100 or (b) when the number of deaths exceeded 10, according to data gathered and made available by Johns Hopkins University [6]. (c) The range of $\mathcal{R}_0$ estimated using two variants of our SEIR model (violet solid and dashed curves) for the range of $\mathcal{T}_d$ estimated in (a) and (b). Vertical lines in the yellow area are $\mathcal{T}_d$ estimates based on the cumulative number cases (orange, from (a)) or the cumulative number of deaths (brown, from (b)). Blue and green solid curves correspond to $\mathcal{R}_0(\mathcal{T}_d)$ according to SEIR models structured and parametrized as in the study of Kucharski et al. [9] ($m = 2$, $n = 2$) and Wu et al. [10] ($m = 1$, $n = 1$).

for China lies in the range 4.7 to 6.6, overlapping with our estimated range for China: 5.6–7.3. The models having one or two 'exposed' substates, often used to estimate the value of $\mathcal{R}_0$, substantially underestimated $\mathcal{R}_0$, cf. figure 1c and the articles by Wearing et al. [12], Wallinga & Lipsitch [13] and Kochańczyk et al. [14].

There are two main reasons why our estimates of the basic reproduction number are higher compared to other published estimates:

(i) Our SEIR model comprises six 'exposed' substates to account for the latent period distribution. As shown in figure 1c, the broader latent period distribution, exponential (i.e. Erlang with $m = 1$), results in lower $\mathcal{R}_0$ estimates than the Erlang with $m = 2$ (at the same remaining model parameters). We characterized sensitivity of $\mathcal{R}_0$ with respect to the mean latent period, $1/\sigma$, in electronic supplementary material, figure S1, while in figure S2 we show that the assumed latent period distribution is in agreement with epidemiological estimates [7,15,16].

(ii) We estimated the doubling time, $\mathcal{T}_d$, from the growth of the number of cumulative cases and cumulative deaths in the two-week-long exponential phases of the epidemic in six locations, obtaining $\mathcal{T}_d$ ranging from 1.86 to 2.96. These values are much lower than the values reported

**Table 1.** Relation between $\mathcal{T}_d$ model parameters (mean latent period or mean incubation period, $1/\sigma$; mean period of infectiousness, $1/\gamma$; and consequent mean generation interval, $\langle GI \rangle$), mean serial period, $\langle SI \rangle$ and $\mathcal{R}_0$. All estimates are based on the epidemic development in Hubei province of China. The unit of all values, except for $\mathcal{R}_0$, is day. Confidence intervals are given in oval brackets; a credible interval is given in square brackets.

| $\mathcal{T}_d$ | $1/\sigma$ | $1/\gamma$ | $\langle SI \rangle$ or $\langle GI \rangle$ | $\mathcal{R}_0$ | reference |
|---|---|---|---|---|---|
| ? | 5.2 | 2.9 | 6.65[a] | 2.35 (1.15–4.77) | Kucharski et al. [9] |
| 5.2 (4.6–6.1) | 6.5 | ? | 7.0 (5.8–8.1) | 1.94 (1.83–2.06) | Wu et al. [17] |
| 6.4 [5.8–7.1] | 6 | 2.4[b] | 8.4 | 2.68 (2.47–2.86) | Wu et al. [10] |
| 7.4 | 5.2 (4.1–7.0) | ? | 7.5 (5.3–19) | 2.2 (1.4–3.9) | Li et al. [16] |

[a]The $\langle GI \rangle$ value is not given in the article but calculated from the assumed values of $1/\sigma$ and $1/\gamma$ as $\langle GI \rangle = 1/\sigma + \frac{1}{2}/\gamma$ [18].
[b]The value $1/\gamma$ was obtained by the authors as $\langle SI \rangle - 1/\sigma$, which is inconsistent with the assumption that the infection occurs in a random time during the period of infectiousness.

in the early influential studies of Wu et al. [10,17] and Li et al. [16]: 5.2 days, 6.4 days and 7.4 days, correspondingly. In these studies, the basic reproduction number has been estimated to lie in between 1.94 and 2.68. A summary in table 1 shows that the lower $\mathcal{R}_0$ estimates follow from much longer estimates of $\mathcal{T}_d$.

## 2.3. Impact of super-spreading on $\mathcal{T}_d$ estimation

The discrepancy in $\mathcal{T}_d$ estimation may be potentially attributed to the fact that not all 'removed' individuals are registered. In the case when the ratio of registered to 'removed' individuals is increasing over time, the true increase of the 'removed' cases may be overestimated. We do not rule out this possibility, although we consider it implausible as the expansion of testing capacity in considered countries has been slower than the progression of the outbreak. We rather attribute the discrepancy to the fact that in the early phase, in which the doubling time (growth rate) is estimated based on individual case reports, the consequences of potential super-spreading events (such as football matches, carnival fests, demonstrations, masses or hospital-acquired infections) are negligible due to a low probability of such events when the number of infected individuals is low. In a given region or country, occurrence of first super-spreading events triggers transition to the faster-exponential growth, in which subsequent super-spreading events become statistically significant and may become decisive drivers of the epidemic spread [19]. Based on case reports in China, Sanche et al. [11] inferred that the initial epidemic period in Wuhan has been dominated by simple transmission chains. Phylogenetic analyses by Worobey et al. [20] revealed that the first cases recorded in USA and Europe did not initiate sustained SARS-CoV-2 transmission networks. In turn, super-spreading events were very likely the main drivers of the epidemic spread in e.g. Italy and Germany, where, in the early exponential phase, spatial heterogeneity of registered cases had been evident [21,22]. In Italy, Spain and France, this explosive phase was followed by a phase of slower growth, during which mass gatherings were forbidden, but quarantine (that finally brought the effective reproduction number below 1) had not been yet introduced.

Motivated by these considerations, we analysed the impact of super-spreading on estimation of $\mathcal{T}_d$ based on stochastic simulations of SEIR model dynamics (see electronic supplementary material, listing S1). Simulations were performed in the perfectly mixed regime according to the Gillespie algorithm [23]. We assumed that a predefined fixed proportion of individuals (equal to 33%, 10%, 3% or 1%) has higher infectiousness and as such is responsible for on average either half of infections (super-spreaders) or two-third of infections (hyper-spreaders). To reproduce these fractions in systems with different assigned proportions of super- or hyper-spreaders, their infectiousness is assumed to be inversely proportional to their ratio in the simulated population. In figure 2, we show dynamics of the epidemic

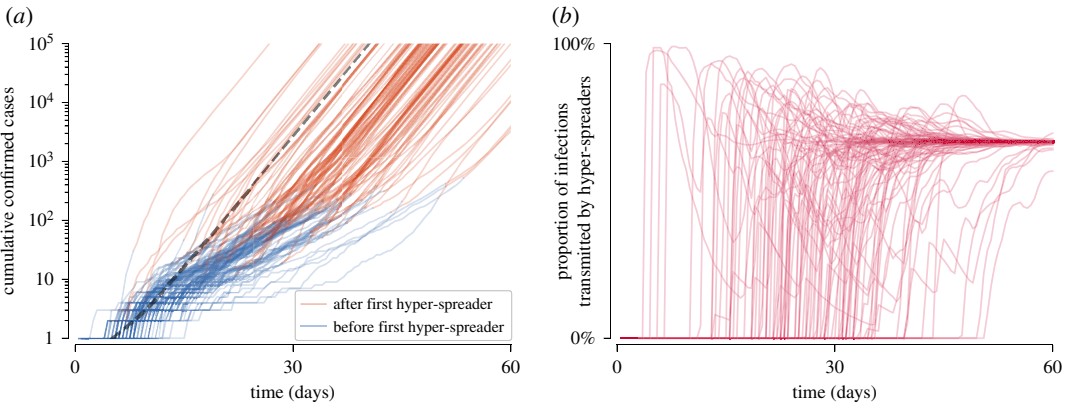

**Figure 2.** Stochastic epidemic spread in the presence of 1% of hyper-spreaders. (a) Trajectories of confirmed cases (cumulative R in terms of SEIR compartments) resulting from 100 independent stochastic simulations. When the first hyper-spreading event occurs, the colour of the line is changed from blue to brown. Dashed grey line shows a deterministic trajectory. (b) Proportion of infections transmitted by hyper-spreaders among all transmission events over time. Stochastic trajectories stabilize at 66.7%. Trajectories shown in both panels results from the same set of simulations; simulations resulting in outbreak failure were discarded. Model parameters used for simulations in both panels: $(m, n) = (6, 1)$, $(1/\sigma, 1/\gamma) = (5.28 \text{ days}, 2.9 \text{ days})$. Infection rate coefficient of hyper-spreaders was set $\beta_h = 198 \times \beta_n$ (where $\beta_n$ is the infection rate coefficient for normal spreaders), which assures that in the deterministic limit 66.7% of infections are transmitted by hyper-spreaders. In turn, $\beta_n$ was set such that the average infection rate coefficient $\beta = 2.97 \times \beta_n$ gives $\mathcal{T}_d = 2$ days (see equation (4.7) in Methods).

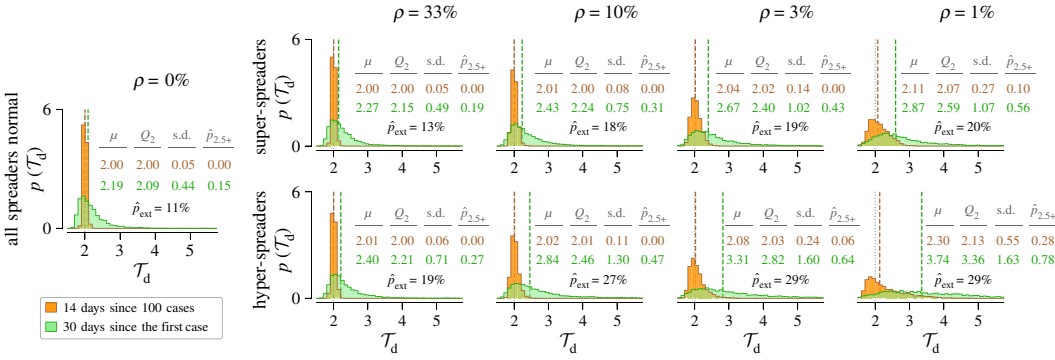

**Figure 3.** Estimation of the doubling time $\mathcal{T}_d$ based on stochastic simulations of the SEIR model with super- and hyper-spreaders. Histograms show probability density $p(\mathcal{T}_d)$ estimated using the '14 days since 100 cases' method (orange) and the '30 days since the first case' method (green). In each column, $\rho$ denotes a fixed proportion of super-spreaders (top row) or hyper-spreaders (bottom row) in the population. For decreasing proportions of super- and hyper-spreaders (from left, except the shared leftmost panel with $\rho = 0$, to right), their infection rate coefficient $\beta$ has been reduced to give the same deterministic $\mathcal{T}_d = 2$ days (vertical dotted grey lines). Remaining model parameters: $(m, n) = (6, 1)$; $(1/\sigma, 1/\gamma) = (5.28 \text{ days}, 2.9 \text{ days})$. Each histogram results from 5000 stochastic simulations starting from a single infected normal individual; trajectories resulting in outbreak failure were discarded; fraction of trajectories that resulted in epidemic extinction for given conditions is given as $\hat{p}_{ext}$. Each distribution is described in terms of its mean ($\mu$), median ($Q_2$ and vertical dashed lines), standard deviation (s.d.) and the fraction of probability mass for $\mathcal{T}_d > 2.5$ days ($\hat{p}_{2.5+}$).

spread in the presence of 1% of hyper-spreaders to demonstrate that the phase of slower growth is transformed into the faster-exponential growth phase upon the occurrence of hyper-spreading events.

We estimated $\mathcal{T}_d$ in two ways: based on one month of growth of the number of new cases since the first registered case (30 days since the first case) and based on growth of new cases in the two-week period after the number of registered cases exceeds 100 (14 days since 100 cases). As we are interested in the initial phase characterized by exponential growth, we assumed that the susceptible population remains constant. In figure 3, we show histograms of $\mathcal{T}_d$ calculated using either the '14 days since 100 cases' method or the '30 days since the first case' method. One may observe that the histograms calculated using the '30 days since the first case' method are broader than those calculated using the

**Table 2.** Doubling time ($\mathcal{T}_d$) estimates using the '14 days since 100 cases' or the '30 days since the first case' method. For all locations except China, the data gathered and made available by Johns Hopkins University [6] are used. As for the second method, the calculation either starts from the first case or from two first cases (if no date is provided for the first case). In the case of China (for which Johns Hopkins University [6] early data are not available) data from [24] are used.

| | '14 days since 100 cases' | | | '30 days since the first case' | | |
|---|---|---|---|---|---|---|
| location | $\mathcal{T}_d$ | from | up to | $\mathcal{T}_d$ | from | up to |
| China | 2.36 | 548 cases (22 Jan 2020) | 23 707 cases (4 Feb 2020) | 4.43 | 1 case (1 Dec 2019) | 37 cases (30 Dec 2019) |
| Italy | 2.56 | 155 cases (23 Feb 2020) | 5883 cases (7 Mar 2020) | 3.25 | 2 cases (31 Jan 2020) | 1128 cases (29 Feb 2020) |
| Spain | 2.11 | 120 cases (2 Mar 2020) | 7798 cases (15 Mar 2020) | 6.35 | 1 case (1 Feb 2020) | 84 cases (1 Mar 2020) |
| France | 2.61 | 130 cases (1 Mar 2020) | 4496 cases (14 Mar 2020) | 12.30 | 2 cases (24 Jan 2020) | 12 cases (22 Feb 2020) |
| UK | 2.88 | 134 cases (2 Mar 2020) | 3077 cases (15 Mar 2020) | 7.66 | 2 cases (31 Jan 2020) | 61 cases (29 Feb 2020) |
| Germany | 2.56 | 130 cases (1 Mar 2020) | 4585 cases (14 Mar 2020) | 12.56 | 1 case (27 Jan 2020) | 17 cases (25 Feb 2020) |
| Switzerland | 2.86 | 114 cases (5 Mar 2020) | 3028 cases (18 Mar 2020) | 2.34 | 1 case (25 Feb 2020) | 10 897 cases (25 Mar 2020) |
| NY (state) | 1.86 | 106 cases (8 Mar 2020) | 11 727 cases (21 Mar 2020) | 1.92 | 1 case (2 Mar 2020) | 75 833 case (31 Mar 2020) |

'14 days since 100 cases' method, and the width of all histograms increases with increasing infectiousness (which is set inversely proportional to $\rho$). When $\mathcal{T}_d$ is calculated using the '14 days since 100 cases' method, its median value is slightly larger than $\mathcal{T}_d$ in the deterministic model (equal to 2 days); however, when $\mathcal{T}_d$ is calculated using the '30 days since the first case' method, then for high infectiousness of super- and hyper-spreaders (correspondingly, for low $\rho$) its median value becomes much larger than the deterministic $\mathcal{T}_d$. Using the '30 days since the first case' method for the case of the lowest considered $\rho = 1\%$, when super-spreaders (hyper-spreaders) have their infectiousness about 100 times (200 times) higher than the infectiousness of normal individuals, one obtains median $\mathcal{T}_d$ larger than $\mathcal{T}_d$ obtained in the deterministic model by 29% (67%), while for '14 days since 100 cases' the $\mathcal{T}_d$ overestimation is negligible, 3% (6%). This difference is caused by low probability of appearance of super- or hyper-spreaders in the first weeks of the outbreak.

We note that $\mathcal{T}_d$ estimation for a given country based on available data is equivalent to the analysis of a single stochastic trajectory and that at a very initial stage the epidemic can cease. Probability of extinction is larger when a small fraction of super-spreaders is responsible for a large fraction of cases. In figure 3, we provide extinction probability, $\hat{p}_{ext}$, which in the extreme case of 1% of hyper-spreaders reaches 29%, whereas without hyper-spreaders (or super-spreaders) is 11%.

The examples shown in figures 2 and 3 are focused on the case in which the $\mathcal{T}_d = 2$ days, which is close to $\mathcal{T}_d$ estimated for Spain and New York State. After removing super-spreaders (assumed to be responsible for 50% of transmissions) the doubling time would be equal to 3.05 days, whereas after removing hyper-spreaders (responsible for 66.7% of transmissions) the doubling time would be equal to 4.24 days. The doubling times in the range 5.2–7.4, obtained by analysing early onsets of the epidemic (Wu *et al.* [10,15] and Li *et al.* [16]), exceed our model prediction obtained after removing 66.7% of transmissions by hyper-spreaders, suggesting that the fraction of transmissions for which hyper-spreaders are responsible can be even larger. Endo *et al.* estimated that 80% of secondary transmissions could have been caused by 10% of infectious individuals [19].

Finally, we compare $\mathcal{T}_d$ estimates obtained for eight considered locations using the '14 days since 100 cases' method (as in figure 1) or the '30 days since the first case' method. As expected, $\mathcal{T}_d$ estimates using '30 days since the first case' method in most cases are larger and more dispersed (in range 1.92–12.6) than

the estimates based on the primary method (1.86–2.88). Results shown in table 2 clearly indicate that the '14 days since 100 cases' method is more reliable. Its disadvantage lies in the fact that for a given location the $\mathcal{T}_d$ estimate is possible when the epidemic is fully developed (see third and fourth columns of table 2). It is, however, important to note that for China the '14 days since 100 cases' method estimate (chosen in our study and giving $\mathcal{R}_0(\mathcal{T}_d)$ in range 5.6–7.3) was possible on 4 February 2020, before the surge of the epidemic in Europe and USA, and more than one month ahead of the first European country-wide lockdown that was imposed in Italy (9 March 2020).

## 3. Conclusion

Based on epidemic data from China, New York State and six European countries, we have estimated that the basic reproduction number $\mathcal{R}_0$ lies in the range 4.7–11.4 (5.6–7.3 for China), which is higher than most previous estimates [4,5,8]. There are two sources of the discrepancy in $\mathcal{R}_0$ estimation. First, in agreement with data on the incubation period distribution (assumed to be the same as the latent period distribution), we used a model with six 'exposed' states, which substantially increases $\mathcal{R}_0$ ($\mathcal{T}_d$) with respect to the models with one or two 'exposed' states. Second, we estimated $\mathcal{T}_d$ based on the two-week period of the exponential growth phase beginning on the day in which the number of cumulative registered cases exceeds 100, or when the number of cumulative registered fatalities exceeds 10. Importantly, values of $\mathcal{T}_d$ estimated from the growth of registered cases and from the growth of the registered fatalities led to similar $\mathcal{R}_0$ estimates. This approach, in contrast to estimation of $\mathcal{R}_0$ based on individual case reports, allows to implicitly take into account super-spreading events that substantially shorten $\mathcal{T}_d$. Spatial heterogeneity of the epidemic spread observed in many European countries, including Italy, Spain and Germany, can be associated with larger or smaller super-spreading events that initiated outbreaks in particular regions of these countries. Lack of, or sporadic super-spreading events in first phase of epidemic explains why $\mathcal{T}_d$ estimated using the '30 days since the first case' method gives in most cases larger and more dispersed values than our method of choice: '14 days since 100 cases'. This, in turn, suggests that in general the reproduction number calculated based on early epidemic development can be probably underestimated, and thus in the case of future epidemics must be considered with caution.

Our estimates are consistent with current epidemic data in Italy, Spain and France. As of 24 April 2020, these countries managed to terminate the exponential growth phase by means of country-wide quarantine. Current COVID-19 Community Mobility Reports [25] show about 80% reduction of mobility in retail and recreation, transit stations and workplaces in these countries. Together with increased social distancing, this reduction possibly lowered the infection rate $\beta$ at least fivefold; additionally, massive testing reduced the infectious period, $1/\gamma$. Consequently, we suspect that the reproduction number $R = \beta/\gamma$ was reduced more than fivefold, which brought it to the values somewhat smaller than 1. This suggests that $\mathcal{R}_0$ in these countries could have been larger than 5.

## 4. Methods

### 4.1. SEIR model equations and parametrization

The dynamics of our SEIR model is governed by the following system of ordinary differential equations:

$$\frac{dS}{dt} = -\beta I(t) S(t)/N \tag{4.1}$$

$$\frac{dE_1}{dt} = \beta I(t) S(t)/N - m\,\sigma E_1(t), \tag{4.2}$$

$$\frac{dE_i}{dt} = m\,\sigma E_{i-1}(t) - m\,\sigma E_i(t), \quad 2 \le i \le m, \tag{4.3}$$

$$\frac{dI_1}{dt} = m\,\sigma E_m(t) - n\,\gamma I_1(t), \tag{4.4}$$

$$\frac{dI_j}{dt} = n\,\gamma I_{j-1}(t) - n\,\gamma I_j(t), \quad 2 \le j \le n, \tag{4.5}$$

$$\frac{dR}{dt} = n\,\gamma I_n(t), \tag{4.6}$$

where $N = S(t) + E_1(t) + \ldots + E_m(t) + I_1(t) + \ldots + I_n(t) + R(t)$ is the constant population size, and $I(t) = I_1(t) + \ldots + I_n(t)$ is the size of infectious subpopulation. As $m$ is the number of 'exposed' substates and $n$ is the

number of 'infectious' substates, there are $m + n + 2$ equations in the system. In the early phase of the epidemic, $1 - S(t)/N \ll 1$ and with constant coefficients $\beta$, $\sigma$ and $\gamma$ the growth of $R$ (as well as $E_i$ and $I_j$) is exponential.

An important property of a given SEIR model parametrization is its implied distribution of generation interval (GI), the period between subsequent infection events in a transmission chain. While the expected GI is easily computable from model parameters as $\langle GI \rangle = \sigma^{-1} + \frac{1}{2}\gamma^{-1}$ (the mean period of infectiousness is halved to reflect the assumption that the infection occurs in a random time during the period of infectiousness [17]), it can be hardly estimated based on even detailed epidemiological data. It should be noted that in some sources the formula $\langle GI \rangle = \sigma^{-1} + \gamma^{-1}$ is used (e.g. [26]), in which it is assumed that the infection occurs at the end of the period of infectiousness, not at a random point of this period. GI may be related to the serial interval, SI, the period between the occurrence of symptoms in the infector and the infectee. Although GI and SI may have different distributions, their means are expected to be equal and thus may be directly compared. Our parametrization implies $\langle GI \rangle = 6.73$ days, which is consistent with $\langle GI \rangle$ of the model by Ferguson et al. (6.5 days) [27] and the estimates of $\langle SI \rangle$ by Wu et al. (7.0 days) [15], Ma et al. (6.8 days) [28] or Bi et al. (6.3 days) [29].

The short period of effective infectiousness reflects the assumption that the individuals with confirmed infection are quickly isolated or self-isolated and then cannot infect susceptible individuals. This enabled us to identify the reported increase of confirmed cases with the transfer of the individuals from the (last substate of the) 'infectious' compartment to the 'removed' compartment of the SEIR model. In addition to the currently diseased individuals that remain isolated, the 'removed' compartment contains the recovered (and assumed to be resistant) and the deceased individuals.

We assume the same $1/\gamma = 2.9$ days in all locations and times, being, however, aware that the mean infectious period may shorten over time due to the implementation of protective health-care practices, increased diagnostic capacity, and contact tracing [29]. In turn, the mean latent period, $1/\sigma$, may be considered an intrinsic property of the disease. As the distribution of the latent period is not known, as a simplification, in our model, the distribution of the latent period (time since infection during which an infected individual cannot infect) is assumed to be the same as the distribution of the incubation period (time since infection during which an infected individual has not yet developed symptoms). We demonstrated the influence of $1/\sigma$ on the estimation of $\mathcal{R}_0$ in electronic supplementary material, figure S1.

## 4.2. Estimation of the doubling time and the basic reproduction number

Growth rates used for estimation of respective doubling times, $\mathcal{T}_d$, were determined by linear regression of the logarithm of the cumulative confirmed cases and cumulative deaths in the exponential phase of the epidemic separately in each of eight considered location. We discarded initial parts of trajectories with less than 100 confirmed cases (or 10 registered fatalities) and used two-week-long periods to strike a balance between: (i) analysis of epidemic progression when stochastic effects associated with individual transmission events, including super-spreading, are relatively small (see stochastic simulation trajectories in electronic supplementary material, figure 2a), and (ii) analysis of the exponential phase of epidemic progression, which is relatively short due to imposition of restrictions. We expect that the trajectories of deaths may be less affected by under-reporting; nevertheless, doubling times obtained from growth rates of cumulative cases and cumulative deaths turn out to be quite consistent.

In the context of our SEIR model, the doubling time $\mathcal{T}_d$ and parameters $\beta, \sigma, \gamma, n, m$ satisfy the relation

$$\beta(\mathcal{T}_d;\ \sigma,\ \gamma,\ m,\ n) = \frac{\frac{\log 2}{\mathcal{T}_d}\left(\frac{\log 2}{\mathcal{T}_d\, m\, \sigma} + 1\right)^m}{1 - \left(\frac{\log 2}{\mathcal{T}_d\, n\, \gamma} + 1\right)^{-n}} \tag{4.7}$$

that enables calculation of the basic reproduction number using the doubling time $\mathcal{T}_d$ estimated directly from the epidemic data as

$$\mathcal{R}_0(\mathcal{T}_d) = \frac{\beta(\mathcal{T}_d;\ \sigma,\ \gamma,\ m,\ n)}{\gamma}, \tag{4.8}$$

in accordance with Wearing et al. [12] and Wallinga & Lipsitch [13].

Data accessibility. All data used in this theoretical study are referenced.

Authors' contributions. M.K. conceived study, performed model and data analysis, prepared figures and wrote manuscript; F.G. conceived study, performed model analysis and prepared figures; T.L. conceived study and wrote manuscript. All authors gave final approval for publication.

Competing interests. We declare we have no competing interest.

Funding. This study was supported by the National Science Centre (Poland) grant no. 2018/29/B/NZ2/00668.

Acknowledgements. We thank the reviewers whose comments helped us to improve the manuscript.

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
