## [Reviewer comments · Royal Society Open Science]

Review History

RSOS-200786.R0 (Original submission)

Review form: Reviewer 1

Is the manuscript scientifically sound in its present form?

No

Are the interpretations and conclusions justified by the results?

No

Is the language acceptable?

Yes

Do you have any ethical concerns with this paper?

No

Have you any concerns about statistical analyses in this paper?

No

Recommendation?

Reject

Comments to the Author(s)

The authors used SEIR model and doubling time estimation to claim that the existing estimates of R_0 are underestimates because the early data did not include superspreaders. The existence of superspreaders and their potential effects on estimation is an important topic, however, there were some critical flaws in their approach which makes the authors' conclusion unreliable. Also, their conclusion that superspreaders led to the underestimation of the existing R_0 estimates is only hypothetical and do not have support from data.

- The authors confuse the interpretations of some key parameters. In the doubling time estimation they used incubation period and infectious period to obtain the generation time distribution, which, in their method, gives about 8-10 days of mean generation time. However, this is clearly longer than the reported mean serial intervals (See Zhao et al, Nishiura et al., He et al. etc.). The incubation period of COVID-19 is estimated to be longer than the latent period as transmission may happen before the onset of symptoms. I suspect this was why the authors' estimate was larger than the existing R_0 estimates, on which now we have substantial literature. Moreover, the SEIR model settings where the authors equate the confirmed case and the removed case does not reflect the actual epidemiology; the case detection (or reporting) and the loss of infectiousness are independent events.

- The authors did not provide how their assumptions about superspreaders can be justified from empirical data. Especially, the scenarios under which 1-3% of the super/hyperspreaders produce more than half of secondary transmissions are extreme and I feel such possibilities are not supported by the existing data. The authors discarded the initial phase of the epidemic data when estimating doubling time to account for superspreaders, but do these initial data points support the authors' hypothesis (ie. was the initial growth when superspreaders might not have arisen consistent with the reproduction rate of non-superspreaders)?

- In the Figure 3 legend the authors state that they discarded trajectories resulting in outbreak failure (extinction?), but did not report what proportion such trajectories accounted for in the overall simulations. I suspect that this may have been the source of bias in their results. Moreover, a high degree of overdispersion (superspreading/hyperspreading) suggests the epidemic is likely to go extinct; although such phenomenon was possible for COVID-19 as some countries took longer from observing the first few imported cases to confirm local epidemic, I am not sure if the authors extreme super/hyperspreaders settings can explain the current global spread of the disease without extinction.

Review form: Reviewer 2

Is the manuscript scientifically sound in its present form?

Yes

Are the interpretations and conclusions justified by the results?

Yes

Is the language acceptable?

Yes

Do you have any ethical concerns with this paper?

No

Have you any concerns about statistical analyses in this paper?

Yes

Recommendation?

Accept with minor revision (please list in comments)

Comments to the Author(s)

The manuscript# RSOS-200786 entitled “Super-spreading events initiated the exponential growth phase of COVID-19 with R_0 higher than initially estimated” is the estimation of reproduction number R_0 value based on superspreading events that lead to the exponential growth of infected population in multiple countries (China, USA and six European nations). Authors used epidemic data from Johns Hopkins University through an interactive web-based dashboard to track COVID-19 in real time and SEIR was used for R_0 calculation. By considering the phase in which the number of registered cases begins to exceed 100, authors show that in the presence of super-spreaders the median trajectory starting from a single infection grows much slower than the average trajectory, which leads to overestimation of the doubling time. Authors used a susceptible–exposed–infected–removed (SEIR) model with six subsequent exposed states to reproduce the shape of the reported distribution of the incubation period and determined a plausible range of R_0 at 4.4–11.7, which is considerably higher than the initially reported estimates.

Authors should define the super spreader and hyper spreader for the readers.

Authors must provide clear reasons for the higher estimated R_0 value of 4.4-11.7 compared to other published studies.

Review form: Reviewer 3 (João Sequeira)

Is the manuscript scientifically sound in its present form?

Yes

Are the interpretations and conclusions justified by the results?

Yes

Is the language acceptable?

Yes

Do you have any ethical concerns with this paper?

No

Have you any concerns about statistical analyses in this paper?

No

Recommendation?

Accept with minor revision (please list in comments)

Comments to the Author(s)

See attached files (Appendix A).

Decision letter (RSOS-200786.R0)

Dear Professor Lipniacki,

The editors assigned to your paper ("Super-spreading events initiated the exponential growth phase of COVID-19 with R_0 higher than initially estimated") have now received comments from reviewers. We would like you to revise your paper in accordance with the referee and Associate Editor suggestions which can be found below (not including confidential reports to the Editor). Please note this decision does not guarantee eventual acceptance.

Please submit a copy of your revised paper before 23-Jul-2020. Please note that the revision deadline will expire at 00.00am on this date. If we do not hear from you within this time then it will be assumed that the paper has been withdrawn. In exceptional circumstances, extensions may be possible if agreed with the Editorial Office in advance. We do not allow multiple rounds of revision so we urge you to make every effort to fully address all of the comments at this stage. If deemed necessary by the Editors, your manuscript will be sent back to one or more of the original reviewers for assessment. If the original reviewers are not available, we may invite new reviewers.

- Data accessibility

<http://datadryad.org/submit?journalID=RSOS&manu=RSOS-200786>

- Competing interests

- Authors' contributions

- Acknowledgements

- Funding statement

on behalf of Professor Tim Rogers (Associate Editor) and Mark Chaplain (Subject Editor)
openscience@royalsociety.org

Associate Editor's comments (Professor Tim Rogers):

Comments to the Author:

When making your major revisions, please pay particular attention to the comments of referee 1, which question the soundness of your methods and the extent to which the data support your conclusions.

Comments to Author:

Reviewers' Comments to Author:
Reviewer: 1

Comments to the Author(s)

The authors used SEIR model and doubling time estimation to claim that the existing estimates of R_0 are underestimates because the early data did not include superspreaders. The existence of superspreaders and their potential effects on estimation is an important topic, however, there

were some critical flaws in their approach which makes the authors' conclusion unreliable. Also, their conclusion that superspreaders led to the underestimation of the existing R_0 estimates is only hypothetical and do not have support from data.

- The authors confuse the interpretations of some key parameters. In the doubling time estimation they used incubation period and infectious period to obtain the generation time distribution, which, in their method, gives about 8-10 days of mean generation time. However, this is clearly longer than the reported mean serial intervals (See Zhao et al, Nishiura et al., He et al. etc.). The incubation period of COVID-19 is estimated to be longer than the latent period as transmission may happen before the onset of symptoms. I suspect this was why the authors' estimate was larger than the existing R_0 estimates, on which now we have substantial literature. Moreover, the SEIR model settings where the authors equate the confirmed case and the removed case does not reflect the actual epidemiology; the case detection (or reporting) and the loss of infectiousness are independent events.

- The authors did not provide how their assumptions about superspreaders can be justified from empirical data. Especially, the scenarios under which 1-3% of the super/hyperspreaders produce more than half of secondary transmissions are extreme and I feel such possibilities are not supported by the existing data. The authors discarded the initial phase of the epidemic data when estimating doubling time to account for superspreaders, but do these initial data points support the authors' hypothesis (ie. was the initial growth when superspreaders might not have arisen consistent with the reproduction rate of non-superspreaders)?

- In the Figure 3 legend the authors state that they discarded trajectories resulting in outbreak failure (extinction?), but did not report what proportion such trajectories accounted for in the overall simulations. I suspect that this may have been the source of bias in their results. Moreover, a high degree of overdispersion (superspreading/hyperspreading) suggests the epidemic is likely to go extinct; although such phenomenon was possible for COVID-19 as some countries took longer from observing the first few imported cases to confirm local epidemic, I am not sure if the authors extreme super/hyperspreaders settings can explain the current global spread of the disease without extinction.

Reviewer: 2

Comments to the Author(s)

The manuscript# RSOS-200786 entitled "Super-spreading events initiated the exponential growth phase of COVID-19 with R_0 higher than initially estimated" is the estimation of reproduction number R_0 value based on superspreading events that lead to the exponential growth of infected population in multiple countries (China, USA and six European nations). Authors used epidemic data from Johns Hopkins University through an interactive web-based dashboard to track COVID-19 in real time and SEIR was used for R_0 calculation. By considering the phase in which the number of registered cases begins to exceed 100, authors show that in the presence of super-spreaders the median trajectory starting from a single infection grows much slower than the average trajectory, which leads to overestimation of the doubling time. Authors used a susceptible-exposed-infected-removed (SEIR) model with six subsequent exposed states to reproduce the shape of the reported distribution of the incubation period and determined a plausible range of R_0 at 4.4-11.7, which is considerably higher than the initially reported estimates.

Authors should define the super spreader and hyper spreader for the readers.

Authors must provide clear reasons for the higher estimated R_0 value of 4.4-11.7 compared to other published studies.

Reviewer: 3

Comments to the Author(s)

See attached files (RSOS_200786.pdf)

Author's Response to Decision Letter for (RSOS-200786.R0)

See Appendix B.

RSOS-200786.R1 (Revision)

Review form: Reviewer 1

Is the manuscript scientifically sound in its present form?

Yes

Are the interpretations and conclusions justified by the results?

No

Is the language acceptable?

Yes

Do you have any ethical concerns with this paper?

No

Have you any concerns about statistical analyses in this paper?

No

Recommendation?

Major revision is needed (please make suggestions in comments)

Comments to the Author(s)

I appreciate the authors' effort to improve the readability of the paper and also clarification regarding my previous comments. However, I still have a few concerns which I suggest the authors consider. I think these are important to render the paper useful and interpretable.

As I understand, the main messages of this paper are the following:

- (i) The basic reproduction number of SARS-CoV-2 is estimated to be 4-11.
 - (ii) This is higher than existing estimates because of 6 exposed states and doubling time used was shorter than previously used.
 - (iii) A possible reason for the shorter estimates of doubling time is superspreading events absent in the earlier phase; simulation shows a change in epidemic growth after the emergence of the first super/hyper spreader.
 - (iv) The authors' approach using 14-day-case counts from the day the cumulative number of cases exceeds 100 is more reliable in estimating R_0 in the presence of superspreading.
- Below I detail my comments for each point.

(i) Now we know that R_0 of SARS-CoV-2 substantially varies between countries, and also even between studies on the same country depending on data and methods used (see Rahman et al. Rev Med Virol, for example). Some already reported R_0 estimates of 4-6 in some countries (although whether these are reliable remains an open question). Given that the authors apparently picked up countries with fastest epidemic growth (many other countries had a slower initial growth), I wonder if treating the value of 4-11 to be R_0 in a general context without specifying the country is a good idea.

(ii) It is nice to have Table R1, but these studies are looking at Wuhan/Hubei with only a limited number of initial cases (therefore inevitably suffer small sample size). Country/region names should be specified to avoid misunderstanding, and also these T_d values should be only

compared against estimates for the same country/region. Even if there was no such existing study for a country, I think the authors could use JHU data before the periods they analyzed in this paper. Indeed, it would be very interesting to see if there was a change in growth rates in the early phase of an epidemic as suggested in Fig 2, which will add empirical evidence to the authors' argument.

(iii) - I concur with Reviewer 3 pointing out that $m = 6$ and $n = 2$ are arbitrary and should be justified. For example, gamma distribution was used to estimate incubation period in some study, and could the authors confirm that the shape of distributions in these existing studies are similar to the one used here, or at least the mean and standard deviation are in line?

- Emergence of hyperspreader: As I mentioned earlier, one way to add support to this hypothesis is to compare this with actual data in the initial phase. Also as in my initial comment, 1% of hyper spreaders who is 200 times infectious than average in Fig 2 is quite extreme and I am not sure if this phenomenon could happen in reality. In the response the authors agreed that the data on superspreading is still limited and thus a range of values needs to be tested. How about more realistic scenarios, such as moderate proportion and infectiousness of superspreaders and/or non-binary heterogeneity represented by a negative binomial distribution with small overdispersion parameter, which now is empirically observed worldwide (for example, Ramanan et al., Sun et al. both in medRxiv)?

(iv) I agree that data in the later period is more reliable to estimate R_0 (especially when there is superspreading). However, the comparison here seems unfair to me because the sample size and time needed to produce an estimate is different between the two approaches. Using initial data is more prone to bias and estimation error, but it may have to be accepted as a necessary trade-off in the earliest analysis. If we have to wait 14 days after the first 100 cases, with a doubling time of 2, the total number of cases could go up to $> 10,000$ which is too late to take action for epidemic control (and if intervention comes in place the estimated reproduction number is out of date). For a fair comparison, I suggest the authors show the difference in time of estimation and the number of cases accumulated during this time gap and discuss the strengths and weaknesses of each approach. Also note that information on the degree of superspreading is extremely difficult in the initial phase when R_0 estimates are most needed.

Minor comment

- Fig 1: Month names should be in English

Review form: Reviewer 2

Is the manuscript scientifically sound in its present form?

Yes

Are the interpretations and conclusions justified by the results?

Yes

Is the language acceptable?

Yes

Do you have any ethical concerns with this paper?

No

Have you any concerns about statistical analyses in this paper?

No

Recommendation?

Accept as is

Comments to the Author(s)

Authors have made the changes based on the reviewers comments. I have no further comments.

Review form: Reviewer 3 (João Sequeira)**Is the manuscript scientifically sound in its present form?**

Yes

Are the interpretations and conclusions justified by the results?

Yes

Is the language acceptable?

Yes

Do you have any ethical concerns with this paper?

No

Have you any concerns about statistical analyses in this paper?

No

Recommendation?

Accept with minor revision (please list in comments)

Comments to the Author(s)

The authors made substantial and adequate improvements.

We briefly highlight important points addressed in the revision process:

1. Elimination of the analysis based upon the three-week interval exponential phase - simplification of data resulting in more homogeneous results.
2. The use of New York state data in substitution of overall USA heterogeneous results. It is an adequate improvement.
3. Estimation of T_{d} from the increase in cumulative cases and cumulative deaths. The under-reporting of deaths should, in theory, be less significant.
4. Figure 1 is clearer.
- 5 The inclusion of generation and serial intervals analysis is an adequate improvement.
6. Proper explanations were given, with the adequate detail, in the Methods and Results sections, concerning our initial doubts and requests.
7. Minor suggestions were included in the attached file (Appendix C).

Decision letter (RSOS-200786.R1)

Dear Professor Lipniacki

On behalf of the Editors, we are pleased to inform you that your Manuscript RSOS-200786.R1 "Super-spreading events initiated the exponential growth phase of COVID-19 with R_0 higher than initially estimated" has been accepted for publication in Royal Society Open Science subject

to minor revision in accordance with the referees' reports. Please find the referees' comments along with any feedback from the Editors below my signature.

Please submit your revised manuscript and required files (see below) no later than 7 days from today's (ie 21-Aug-2020) date. Note: the ScholarOne system will 'lock' if submission of the revision is attempted 7 or more days after the deadline. If you do not think you will be able to meet this deadline please contact the editorial office immediately.

on behalf of Professor Tim Rogers (Associate Editor) and Mark Chaplain (Subject Editor)
openscience@royalsociety.org

Associate Editor Comments to Author (Professor Tim Rogers):
Comments to the Author:
Revisions addressing the concerns of referee 3 are necessary before publication.

Reviewer comments to Author:
Reviewer: 2

Comments to the Author(s)
Authors have made the changes based on the reviewers comments. I have no further comments.

Reviewer: 3

Comments to the Author(s)
The authors made substantial and adequate improvements.

We briefly highlight important points addressed in the revision process:

1. Elimination of the analysis based upon the three-week interval exponential phase - simplification of data resulting in more homogeneous results.
2. The use of New York state data in substitution of overall USA heterogeneous results. It is an adequate improvement.

3. Estimation of $T_{\{d\}}$ from the increase in cumulative cases and cumulative deaths. The under-reporting of deaths should, in theory, be less significant.
4. Figure 1 is clearer.
- 5 The inclusion of generation and serial intervals analysis is an adequate improvement.
6. Proper explanations were given, with the adequate detail, in the Methods and Results sections, concerning our initial doubts and requests.
7. Minor suggestions were included in the attached file.

Reviewer: 1

Comments to the Author(s)

I appreciate the authors' effort to improve the readability of the paper and also clarification regarding my previous comments. However, I still have a few concerns which I suggest the authors consider. I think these are important to render the paper useful and interpretable.

As I understand, the main messages of this paper are the following:

- (i) The basic reproduction number of SARS-CoV-2 is estimated to be 4-11.
 - (ii) This is higher than existing estimates because of 6 exposed states and doubling time used was shorter than previously used.
 - (iii) A possible reason for the shorter estimates of doubling time is superspreading events absent in the earlier phase; simulation shows a change in epidemic growth after the emergence of the first super/hyper spreader.
 - (iv) The authors' approach using 14-day-case counts from the day the cumulative number of cases exceeds 100 is more reliable in estimating R_0 in the presence of superspreading.
- Below I detail my comments for each point.

(i) Now we know that R_0 of SARS-CoV-2 substantially varies between countries, and also even between studies on the same country depending on data and methods used (see Rahman et al. Rev Med Virol, for example). Some already reported R_0 estimates of 4-6 in some countries (although whether these are reliable remains an open question). Given that the authors apparently picked up countries with fastest epidemic growth (many other countries had a slower initial growth), I wonder if treating the value of 4-11 to be R_0 in a general context without specifying the country is a good idea.

(ii) It is nice to have Table R1, but these studies are looking at Wuhan/Hubei with only a limited number of initial cases (therefore inevitably suffer small sample size). Country/region names should be specified to avoid misunderstanding, and also these T_d values should be only compared against estimates for the same country/region. Even if there was no such existing study for a country, I think the authors could use JHU data before the periods they analyzed in this paper. Indeed, it would be very interesting to see if there was a change in growth rates in the early phase of an epidemic as suggested in Fig 2, which will add empirical evidence to the authors' argument.

(iii) - I concur with Reviewer 3 pointing out that $m = 6$ and $n = 2$ are arbitrary and should be justified. For example, gamma distribution was used to estimate incubation period in some study, and could the authors confirm that the shape of distributions in these existing studies are similar to the one used here, or at least the mean and standard deviation are in line?

- Emergence of hyperspreader: As I mentioned earlier, one way to add support to this hypothesis is to compare this with actual data in the initial phase. Also as in my initial comment, 1% of hyper spreaders who is 200 times infectious than average in Fig 2 is quite extreme and I am not sure if this phenomenon could happen in reality. In the response the authors agreed that the data on superspreading is still limited and thus a range of values needs to be tested. How about more realistic scenarios, such as moderate proportion and infectiousness of superspreaders and/or non-binary heterogeneity represented by a negative binomial distribution with small overdispersion parameter, which now is empirically observed worldwide (for example, Ramanan et al., Sun et al. both in medRxiv)?

(iv) I agree that data in the later period is more reliable to estimate R_0 (especially when there is superspreading). However, the comparison here seems unfair to me because the sample size and time needed to produce an estimate is different between the two approaches. Using initial data is

more prone to bias and estimation error, but it may have to be accepted as a necessary trade-off in the earliest analysis. If we have to wait 14 days after the first 100 cases, with a doubling time of 2, the total number of cases could go up to > 10,000 which is too late to take action for epidemic control (and if intervention comes in place the estimated reproduction number is out of date). For a fair comparison, I suggest the authors show the difference in time of estimation and the number of cases accumulated during this time gap and discuss the strengths and weaknesses of each approach. Also note that information on the degree of superspreading is extremely difficult in the initial phase when R0 estimates are most needed.

Minor comment

- Fig 1: Month names should be in English

===PREPARING YOUR MANUSCRIPT===

- one version identifying all the changes that have been made (for instance, in coloured highlight, in bold text, or tracked changes);
- a 'clean' version of the new manuscript that incorporates the changes made, but does not highlight them. This version will be used for typesetting.

===PREPARING YOUR REVISION IN SCHOLARONE===

<https://royalsociety.org/journals/authors/author-guidelines/#supplementary-material> to include a suitable title and informative caption. An example of appropriate titling and captioning may be found at https://figshare.com/articles/Table_S2_from_Is_there_a_trade-off_between_peak_performance_and_performance_breadth_across_temperatures_for_aerobic_scops_in_teleost_fishes_/3843624.

Author's Response to Decision Letter for (RSOS-200786.R1)

See Appendix D.

Decision letter (RSOS-200786.R2)

Dear Professor Lipniacki,

It is a pleasure to accept your manuscript entitled "Super-spreading events initiated the exponential growth phase of COVID-19 with R_0 higher than initially estimated" in its current form for publication in Royal Society Open Science.

COVID-19 rapid publication process:

We are taking steps to expedite the publication of research relevant to the pandemic. If you wish, you can opt to have your paper published as soon as it is ready, rather than waiting for it to be published the scheduled Wednesday.

This means your paper will not be included in the weekly media round-up which the Society sends to journalists ahead of publication. However, it will still appear in the COVID-19 Publishing Collection which journalists will be directed to each week (<https://royalsocietypublishing.org/topic/special-collections/novel-coronavirus-outbreak>).

If you wish to have your paper considered for immediate publication, or to discuss further, please notify opencience_proofs@royalsociety.org and press@royalsociety.org when you respond to this email.

You can expect to receive a proof of your article in the near future. Please contact the editorial office (opencience_proofs@royalsociety.org) and the production office (opencience@royalsociety.org) to let us know if you are likely to be away from e-mail contact -- if you are going to be away, please nominate a co-author (if available) to manage the proofing process, and ensure they are copied into your email to the journal.

Kind regards,
Andrew Dunn
Royal Society Open Science Editorial Office
Royal Society Open Science
opencience@royalsociety.org

on behalf of Professor Tim Rogers (Associate Editor) and Mark Chaplain (Subject Editor)
opencience@royalsociety.org

Appendix A

Royal Society Open Science review

Manuscript ID:
RSOS-200786

Title:
Super-spreading events initiated the exponential growth phase of COVID-19 with R_0 higher than initially estimated

1 Overall review

The authors present a SEIR model allowing for a more accurate estimation of R_0 with the presence of disease super-spreading in the current SARS-CoV2 pandemic. Particular attention was drawn to the initial lower disease growth spreading phase.

The present SEIR model includes multiple exposed ($m=6$) and infectious ($n=2$) stages with a gamma-type Erlang distribution of incubation time and infectious time period, respectively. This approach is known to lead to a more realistic (less optimistic) estimation of the basic reproductive number R_0 based upon the empirical determination of the infection doubling time T_d , when compared to exponential-distribution models, or to models with inadequate assumptions about the disease incubation time period. Stochastic simulations of model dynamics were presented, based upon the use of a small proportion of super- and hyper-spreaders in the general population.

The paper addresses an important aspect of COVID-19 pandemic concerning the need for more robust preventive strategies directed at disease super-spreaders in earlier stages of disease transmission, while focusing on the risks of downplaying those preventive interventions due to underestimation of R_0 .

2 Major points to be addressed

The authors use sound statistical definitions throughout the paper. But several of those definitions lack adequate explanation in the text. The rationale

behind the choice of some of the distribution parameters should be presented with enough detail, in order to facilitate the reading task.

a)

In page 2/12, lines 58-60

Why did the authors consider for analysis the epidemic phase in which the number of registered cases began to exceed one hundred? Why one hundred, and not a different cut-off? A short explanation should suffice.

The authors should also address the fact that by comparing countries with different population sizes, that cut-off may induce small differences in proportional disease prevalence rates per 100.000 inhabitants, among different countries.

b)

In page 3/12, line 2

The assumption that super-spreading events are implausible at the beginning of the pandemic is a little tricky.

If we look at what locally happened in Bergamo in February/March of 2020, after the game Atalanta-Valencia of the European Champions League, we see that super-spreading could be an isolated single explosive event before any previous spreading. While the epidemic was in its initial phase in Milano (where the game took place) almost 40.000 Atalanta fans went from Bergamo to Milano to watch the game. The super-spreading event that occurred during and after the game was responsible for the later outbreak in Bergamo, when those fans returned home. The population of Bergamo is estimated to be close to 120.000 inhabitants. So, this flash hyper-spreading event precipitated the Bergamo outbreak by instantly infecting close to a third of the town population. In a similar note, the small contingent of 3.000 Valencia fans, was probably also responsible for the earlier disease outbreak in Valencia and later on in Spain, that followed soon after that game.

So, super-spreading events may not be that implausible at the initial phase of a pandemic. The authors should address this point presenting evidence of their statement in page 3/12, line 2.

c)

In page 3/12, lines 3-7, the following sentence is unclear:

We show that in the presence of super-spreaders the median trajectory starting from a single infection grows much slower than the average trajectory, which very likely leads to overestimation of the doubling time.

The *median trajectory* of what? The infection rate? The absolute number of cases? The infection growth rate?

d)

In page 3/12, line 33-34

The meaning of T_d should be defined at this stage, and not later in the same page, in line 44.

e)

In page 3/12, line 34-42

The authors should present the rationale for the use of a two and three weeks period for estimation of T_d in the Methods section.

f)

In page 3/12, line 35

In the present SEIR model, the authors defined the R stage as **removed**. Although this definition may be partially acceptable, it does not comply with the well accepted definition of stages in the SEIR model in which R stands for **recovered** and not **removed** - see Vynnycky E. and White R G. An introduction to infectious disease modelling, Oxford, 2010.

g)

In page 8/12, lines 26-33

The present model is based on a gamma-type Erlang distribution involving the precise determination of the infection doubling time T_d in several world regions affected by the pandemic (China, USA and Europe). This type of distribution is usually considered more realistic than exponential-distribution based SEIR models.

Exponentially distributed models reveal an initial slower epidemic growth, while predicting a smaller peak number of cases and lasting longer than

gamma-distribution models.

The authors defined the SEIR model with $m = 6$ exposed stages and $n = 2$ infectious stages. These m and n subclasses of exposed and infectious stages, respectively, are usually accepted as efficient computational alternatives for modelling infections with gamma-type probability distributions.

However, models with higher values of m tend to reduce the variance of the incubation period, leading to an overestimation of R_0 .

In the Methods section, the authors should present the rationale for particularly choosing $m = 6$ and $n = 2$ in Erlang distributions for the incubation and infectious period, respectively.

A simple explanation should be adequate.

3 Minor aspects

h)

In page 4/12, lines 41-42

The following sentence is not precise and should be rephrased.

Our calculations suggest the range of R_0 that is higher than initially estimated (Boldog et al., 2020; Liu et al., 2020).

Our suggestion:

*Our calculations suggest the range of R_0 **to be** higher than initially estimated (Boldog et al., 2020; Liu et al., 2020).*

i)

In page 7/12, line 30

In the following sentence, we suggest replacing the word **expect** by a more adequate expression (eventually, the word **suspect** could be used instead; other sound alternatives may also be acceptable)

*Consequently, we **expect** that...*

4 Conclusion

The present paper is scientifically sound, focusing on a relevant aspect of COVID-19 early transmission that should be critical in defining better preventive strategies by public health institutions at country and regional level.

Tomasz Lipniacki
Department of Biosystems and Soft Matter
Institute of Fundamental Technological Research
Polish Academy of Sciences

Pawińskiego 5B, 02-106 Warsaw, Poland
Phone: +48 22 8261280
E-mail: tlipnia@ippt.pan.pl
Website: <http://pmbm.ippt.pan.pl>

Subject: **Revision of manuscript RSOS-200786**

August 3, 2020

Dear Reviewers, Dear Editor,

We revised our manuscript “Super-spreading events initiated the exponential growth phase of COVID-19 with R_0 higher than initially estimated” addressing the raised issues. All essential changes are shown in colour in a separate PDF file. The main modifications introduced during revision are as follows:

1. We have made several adjustments in estimation of doubling time, \mathcal{T}_d :
 - (a) We replaced analysis of the epidemic trajectory of the USA with the analysis of trajectory of New York State. The analysis of the exponential phase in the whole USA is problematic due to the lack of synchrony between states.
 - (b) We estimated \mathcal{T}_d in seven countries and New York State from the increase in the number of cumulative registered cases (Figure 1a) and, additionally, from the cumulative deaths (new panel *b* in Figure 1), obtaining very similar values. We think that this strengthens our analysis as the extent of under-reporting of deaths is expected to be smaller than that of confirmed cases.
 - (c) We analysed only two-week periods of the exponential phase. We decided to abandon the analysis that used three-week periods, because in some countries the epidemic growth in their respective three-week periods is clearly not exponential, biasing initial \mathcal{T}_d estimation required to determine \mathcal{R}_0 .
 - (d) The growth rate used for estimation of \mathcal{T}_d (Figure 1a and Figure 1b) is determined by a simple linear regression of the logarithm of the cumulative confirmed cases and cumulative deaths. Previously, growth rate was determined only based on the growth of the cumulative cases in a considered period. We think that current approach is less sensitive to fluctuations.
2. Following the criticism of the first Referee, in the revised manuscript we fully abandoned the model variant with the latent time of 6.47 days (assumed after Backer et al., 2020, Eurosurveillance 25, 2000062), that had been used as an upper bound in estimation of \mathcal{R}_0 . Now, we consider two slightly different model variants, both assuming the mean latency period of 5.28 days and mean infectious period of 2.9 days, that imply the mean generation interval of 6.73 days. Also, a delay–differential equations-based model with Dirac δ -distributed latent period has been removed.
3. We included a new supplementary figure (Figure S1) in which we show the dependence of the basic reproduction number on the latency period. As stated by the first Referee, the reproduction number is a growing function of the latency period (and of generation interval).

The responses to Referees' specific comments are included below.

We hope that you will find the revised manuscript suitable for publication in the *Royal Society Open Science*.

Sincerely,
Tomasz Lipniacki
Professor and Head

Response to the Report from Reviewer #1

The authors used SEIR model and doubling time estimation to claim that the existing estimates of R_0 are underestimates because the early data did not include superspreaders. The existence of superspreaders and their potential effects on estimation is an important topic, however, there were some critical flaws in their approach which makes the authors' conclusion unreliable. Also, their conclusion that superspreaders led to the underestimation of the existing R_0 estimates is only hypothetical and do not have support from data.

Response: We improved and explained our approach as detailed in the responses to specific comments.

- The authors confuse the interpretations of some key parameters. In the doubling time estimation they used incubation period and infectious period to obtain the generation time distribution, which, in their method, gives about 8–10 days of mean generation time. However, this is clearly longer than the reported mean serial intervals (See Zhao et al., Nishiura et al., He et al. etc.). The incubation period of COVID-19 is estimated to be longer than the latent period as transmission may happen before the onset of symptoms. I suspect this was why the authors' estimate was larger than the existing R_0 estimates, on which now we have substantial literature.

Response: First, let us explain that the doubling time, \mathcal{T}_d , is estimated directly from the empirical increase in the number of cumulative confirmed cases in the two-week-long exponential phase as shown in Figure 1a. In the revised manuscript, we additionally estimated \mathcal{T}_d from the cumulative deaths (new panel b in Figure 1), obtaining very similar values. We expect that the trajectories of deaths are less affected by under-reporting; also, in contrast to confirmed cases-based analysis, the extent of under-reporting may be estimated for a given period based on the “excess deaths” that appear above the expected number of deaths when the officially recorded COVID-19 fatalities are subtracted from the total number of deaths [Tozer & González, Dale & Stylianou]. We have expanded the Methods section with a description of how \mathcal{T}_d was estimated

-
- Tozer J. & González M., *Tracking COVID-19 excess deaths across countries*, <https://www.economist.com/graphic-detail/2020/04/16/tracking-covid-19-excess-deaths-across-countries>, Accessed: 2020-06-26, 2020.
 - Dale B. & Stylianou N., *Coronavirus: what is the true death toll of the pandemic*, <https://www.bbc.com/news/world-53073046>, Accessed: 2020-06-26, 2020.

(new subsection 4(b)), from which it is now clear that the generation time distribution has not been used to estimate \mathcal{T}_d .

We are aware that generation time distribution, determined in SEIR-type models by the assumed latent period and infectious period distributions, crucially affects estimation of \mathcal{R}_0 . In the following two sections we address Reviewer’s concerns regarding generation time distribution and duration of the incubation period, and discuss their relation to the estimated \mathcal{R}_0 .

Mean generation/serial time

In the revised manuscript we fully abandoned the model variant with the incubation time of 6.47 days (assumed after Backer *et al.*) and focused on the model variant in which we assumed that:

- the latent period is equal to the incubation period and is Erlang-distributed with the shape parameter $m = 6$ and the mean of 5.28 days = $1/\sigma$ (after Lauer *et al.*),
- the infectious period is Erlang-distributed with the shape parameter $n = 1$ (exponentially distributed) or $n = 2$, and the mean of 2.9 days = $1/\gamma$ (after Liu *et al.* and as in Kucharski *et al.*).

Correspondingly, the generation time distribution has the mean equal $\langle \text{GI} \rangle = \sigma^{-1} + \frac{1}{2}\gamma^{-1} = \mathbf{6.73 \text{ days}}$ (and the median of 6.28 days for $n = 1$, 6.39 for $n = 2$). In the expression for $\langle \text{GI} \rangle$, the mean period of infectiousness is halved, which reflects the assumption that the infection occurs at a random time during the period of infectiousness — see, e.g., the book *Infectious Disease Epidemiology: Theory and Practice* edited by Nelson & Williams (3rd edition, 2014, Jones & Bartlett Learning, Burlington, MA). Of note, in the literature one may also find the relation $\langle \text{GI} \rangle = \sigma^{-1} + \gamma^{-1}$ (see, e.g. Lipsitch *et al.*), in which it is assumed that the infection occurs at the end of the period of infectiousness, not at a random point of this period. We do not find this assumption reasonable.

The mean generation time implied by our parametrisation is consistent with the estimates by Wu *et al.* (based on 43 infector–infectee pairs from Wuhan, mean serial interval (SI) was estimated as 7.0 days with 95% confidence interval (CI) of 5.8–8.1 days), Ma *et al.* (based on 689 infector–infectee pairs from various countries, mean SI was 6.7 days with 95% CI of 6.31–7.10 days), Ferguson *et al.* (literature-based parametrisation of the model, that included 1/2-day presymptomatic transmission, yielded mean GI of 6.5 days), or Bi *et al.* (based on 48 pairs, mean SI has been estimated as 6.3 days with 95% CI of 5.2–7.6 days).

Regarding the literature references mentioned by the Referee, our mean generation time of 6.73 days borders on the 95% CI of SI estimated by Zhao *et al.* (based on 56 confirmed cases from Hong Kong, in

-
- Backer J. A. *et al.* (2020) *Eurosurveillance* **25**, 2000062, DOI: 10.2807/1560-7917.ES.2020.25.5.2000062.
 - Lauer S. A. *et al.* (2020) *medRxiv*, DOI: 10.1101/2020.02.02.20020016.
 - Liu T. *et al.* (2020) *bioRxiv*, DOI: 10.1101/2020.01.25.919787.
 - Kucharski A. J. *et al.* (2020) *Lancet Infectious Diseases*, DOI: 10.1016/S1473-3099(20)30144-4.
 - Lipsitch M. *et al.* (2003) *Science* **300**, DOI: 10.1126/science.1086616, 1966–1970.
 - Wu J. T. *et al.* (2020^a) *Nature Medicine* **26**, DOI: 10.1038/s41591-020-0822-7.
 - Ma S. *et al.* (2020) *medRxiv*, DOI: 10.1101/2020.03.21.20040329.
 - Ferguson N. *et al.*, Report 9, <https://spiral.imperial.ac.uk:8443/handle/10044/1/77482>, Accessed: 2020-03-26.
 - Bi Q. *et al.* (2020) *The Lancet Infectious Diseases*, DOI: 10.1016/S1473-3099(20)30287-5.
 - Zhao S. *et al.* (2020) *medRxiv*, DOI: 10.1101/2020.02.21.20026559.

part after introduction of the state of emergency, these authors obtained an estimated mean SI of 4.4 days with 95% CI of 2.9–6.7 days) and by He *et al.* (based on 77 infector–infectee pairs from many locations of East Asia, including Hong Kong, SI has been estimated to have a mean of 5.8 days with 95% CI of 4.8–6.8 days). The SI estimate by Nishiura *et al.* is clearly shorter, 4.7 days with 95% credible interval of 3.7–6.0 days, which is close to or shorter than the median incubation period. The estimate is however based on 28 infector–infectee pairs, out of which 12 pairs were family clusters. It is reasonable to expect that a quicker, and possibly pre-symptomatic, transmission may occur in households.

Finally, let us notice that in a recent theoretical study by Peak *et al.*, published in *The Lancet: Infectious Diseases*, two values of generation time are considered in parallel: 4.8 days and 7.5 days.

Incubation period distribution

To address Reviewer’s concerns regarding the impact of the incubation period on estimation of \mathcal{R}_0 , we demonstrated sensitivity of \mathcal{R}_0 with respect to the mean of the latent period (assumed identical to the incubation period) distribution, $1/\sigma$ (see Figure R1; the figure has become Figure S1 in the electronic supplementary materials featuring the revised manuscript). As the mean infectious period, $1/\gamma$, may shorten over time due to the implementation of protective health care practices, increased diagnostic capacity, and contact tracing [Bi *et al.*], parameter $1/\sigma$ may be considered an intrinsic property of the disease.

Figure R1: Basic reproduction number, \mathcal{R}_0 , vs. mean latent period, $1/\sigma$, assuming doubling time $\mathcal{T}_d = 2$ days and SEIR model parameters as indicated in the legend. The default $1/\sigma = 5.28$ days is marked with a dotted vertical line.

Finally, we are aware that there is a substantial body of literature suggesting much lower values of \mathcal{R}_0 . The \mathcal{R}_0 value of uncontrolled epidemic is very important as it determines the extent of measures that should be imposed to limit the spread of epidemic. We think that underestimation of \mathcal{R}_0 was one of the causes of insufficient response to the epidemic in Europe and USA. It should be noticed that the lower \mathcal{R}_0 estimates were not caused by lower estimates of the generation time but by much higher estimates of the doubling time, T_d . Let us refer to five very influential studies summarised in Table R1 (also included in the revised manuscript), in which one can see that the lower \mathcal{R}_0 estimates follow from the much longer estimates of T_d .

- He X. *et al.* (2020) *Nature Medicine* **26**, DOI: 10.1038/s41591-020-0869-5.
- Nishiura H. *et al.* (2020) *International Journal of Infectious Diseases* **93**, DOI: 10.1016/j.ijid.2020.02.060.
- Peak C. M. *et al.* (2020) *The Lancet Infectious Diseases*, DOI: 10.1016/S1473-3099(20)30361-3.
- Bi Q. *et al.* (2020) *The Lancet Infectious Diseases*, DOI: 10.1016/S1473-3099(20)30287-5.

These long T_d estimates however disagree with empirical T_d estimates presented in Figure 1a and Figure 1b.

T_d	$1/\sigma$	$1/\gamma$	$\langle SI \rangle$ or $\langle GI \rangle$	\mathcal{R}_0	Reference
?	5.2	2.9	6.65 ^a	2.35 (1.15–4.77)	Kucharski et al. (600+ citations)
5.2 (4.6–6.1)	6.5	?	7.0 (5.8–8.1)	1.94 (1.83–2.06)	Wu et al.^(a) (400+ citations)
6.4 [5.8–7.1]	6	2.4 ^b	8.4	2.68 (2.47–2.86)	Wu et al.^(b) (1500+ citations)
7.4	5.2 (4.1–7.0)	?	7.5 (5.3–19)	2.2 (1.4–3.9)	Li et al. (4300+ citations)

Table R1: Relation between T_d , model parameters (mean latent period or mean incubation period, $1/\sigma$, mean period of infectiousness, $1/\gamma$, and consequent mean generation interval, $\langle GI \rangle$), mean searial period, $\langle SI \rangle$, and \mathcal{R}_0 . The unit of all values, except for \mathcal{R}_0 , is day. Confidence intervals are given in oval brackets; a credible interval is given in square brackets. Numbers of citations are given according to Google Scholar as of August 1st, 2020. ^aThe $\langle GI \rangle$ value is not given in the article but calculated from the assumed values of $1/\sigma$ and $1/\gamma$ as $\langle GI \rangle = 1/\sigma + \frac{1}{2}/\gamma$. ^bThe value $1/\gamma$ was obtained by the authors as $\langle SI \rangle - 1/\sigma$, which is inconsistent with the assumption that the infection occurs in a random time during the period of infectiousness.

- Moreover, the SEIR model settings where the authors equate the confirmed case and the removed case does not reflect the actual epidemiology; the case detection (or reporting) and the loss of infectiousness are independent events.

Response: The Referee is right in saying that detection or reporting is not equivalent to the loss of infectiousness. We have assumed, however, that the individuals with confirmed infection are isolated or self-isolated and thus cannot infect other susceptible individuals. This assumption is a simplification present in most models, in which the period of effective infectiousness is about 3 days, much shorter than the period of “medical” infectiousness (see, e.g., Kucharski et al.). We have included this explanation in the revised manuscript.

- The authors did not provide how their assumptions about superspreaders can be justified from empirical data. Especially, the scenarios under which 1–3% of the super/hyperspreaders produce more than half of secondary transmissions are extreme and I feel such possibilities are not supported by the existing data. The authors discarded the initial phase of the epidemic data when estimating doubling time to account for superspreaders,

- Kucharski A. J. et al. (2020) *Lancet Infectious Diseases*, DOI: 10.1016/S1473-3099(20)30144-4.
- Wu J. T. et al. (2020^a) *Nature Medicine* **26**, DOI: 10.1038/s41591-020-0822-7.
- Wu J. T. et al. (2020^b) *The Lancet* **395**, DOI: 10.1016/S0140-6736(20)30260-9.
- Li Q. et al. (2020) *New England Journal of Medicine*, DOI: 10.1056/NEJMoa2001316.
- Sanche S. et al. (2020) *Emerging Infectious Diseases* **26**, DOI: 10.3201/eid2607.200282.

but do these initial data points support the authors' hypothesis (ie. was the initial growth when superspreaders might not have arisen consistent with the reproduction rate of non-superspreaders)?

Response: The examples shown in Figure 2 and Figure 3 are focused on the case in which the $\mathcal{T}_d = 2$ days, which is close to \mathcal{T}_d estimated for Spain and N.Y. State. After removing super-spreaders (assumed to be responsible for 50% of transmissions) the doubling time would be equal 3.05 days, whereas after removing hyper-spreaders (responsible for 66.7% of transmissions) the doubling time would be equal to 4.24 days. The doubling times in the range 5.2–7.4, obtained by analysing early onsets of the epidemic (Wu *et al.*^(a), Wu *et al.*^(b), and Li *et al.*), exceed our model prediction obtained after removing 66.7% of transmissions by hyper-spreaders, suggesting that the fraction of transmissions for which hyper-spreaders are responsible can be even larger.

We add this (we think important) information when discussing Figure 3 and are grateful to the Reviewer for posing the question. However, since a reliable estimate of the frequency of super-spreaders (or super-spreading events) is not known, we consider scenarios with various fractions of super-spreaders that can be responsible for two different proportions of transmissions. Super-spreading events — such as the football game Atalanta *vs.* Valencia of the European Champions League (mentioned by the third Referee) or Carnival in Gangelt in Germany, that were responsible for accelerating outbreaks in Lombardy (especially province of Bergamo) and València (Spain), and in North Rhine-Westphalia — suggest that such events were significant catalysers of the epidemic spread in Europe.

- In the Figure 3 legend the authors state that they discarded trajectories resulting in outbreak failure (extinction?), but did not report what proportion such trajectories accounted for in the overall simulations. I suspect that this may have been the source of bias in their results. Moreover, a high degree of overdispersion (superspreading/hyperspreading) suggests the epidemic is likely to go extinct; although such phenomenon was possible for COVID-19 as some countries took longer from observing the first few imported cases to confirm local epidemic, I am not sure if the authors extreme super/hyperspreaders settings can explain the current global spread of the disease without extinction.

Response: The Referee is right in saying that the epidemic (at its very initial stage) can cease, and that the probability of extinction is larger when a small fraction of super-spreaders is responsible for a large fraction of cases. In the revised manuscript we provided extinction probability, which in the extreme case of 1% of hyper-spreaders reaches 29%, whereas without super-spreaders (or hyper-spreaders) is 11% (see updated Figure 3).

Phylogenetic analyses revealed that first cases recorded in USA and Europe did not initiate sustained SARS-CoV-2 transmission networks [Worobey *et al.*]. Although protective measures possibly had an import-

-
- Wu J. T. *et al.* (2020^a) *Nature Medicine* **26**, DOI: 10.1038/s41591-020-0822-7.
 - Wu J. T. *et al.* (2020^b) *The Lancet* **395**, DOI: 10.1016/S0140-6736(20)30260-9.
 - Li Q. *et al.* (2020) *New England Journal of Medicine*, DOI: 10.1056/NEJMoa2001316.
 - Worobey M. *et al.* (2020) *bioRxiv*, DOI: 10.1101/2020.05.21.109322.

ant role in containment of these first failed outbreaks in USA and Europe, it is likely that super-spreading events are critical for seeding successful outbreaks of COVID-19 in new territories [Endo *et al.*].

Response to the Report from Reviewer #2

The manuscript# RSOS-200786 entitled "Super-spreading events initiated the exponential growth phase of COVID-19 with R_0 higher than initially estimated" is the estimation of reproduction number R_0 value based on superspreading events that lead to the exponential growth of infected population in multiple countries (China, USA and six European nations). Authors used epidemic data from Johns Hopkins University through an interactive web-based dashboard to track COVID-19 in real time and SEIR was used for R_0 calculation. By considering the phase in which the number of registered cases begins to exceed 100, authors show that in the presence of super-spreaders the median trajectory starting from a single infection grows much slower than the average trajectory, which leads to overestimation of the doubling time. Authors used a susceptible–exposed–infected–removed (SEIR) model with six subsequent exposed states to reproduce the shape of the reported distribution of the incubation period and determined a plausible range of R_0 at 4.4–11.7, which is considerably higher than the initially reported estimates.

Authors should define the super spreader and hyper spreader for the readers. Authors must provide clear reasons for the higher estimated R_0 value of 4.4–11.7 compared to other published studies.

Response: In our analysis shown in Figure 2 and Figure 3 we considered two model variants in which super-spreaders are responsible for either 50% or 66.7% of infections; in the latter case these super-spreaders are called *hyper*-spreaders. This is explained clearly in the revised manuscript.

There are two main reasons why our estimates of the basic reproduction number are higher compared to other published estimates:

1. Our SEIR model comprises 6 exposed states, which is equivalent to assuming that the latent time (assumed to be identical to the incubation time) is Erlang-distributed with the shape parameter $m = 6$, in agreement with Lauer *et al.* As shown in Figure 1c, broader incubation period distributions, exponential or Erlang with $m = 2$, result in lower \mathcal{R}_0 estimates (at the same remaining model parameters).
2. We estimated the doubling time, \mathcal{T}_d , from the growth of the number of cumulative cases and cumulative deaths in the two-week-long exponential phases of the epidemic in six European countries, N.Y. State, and China (see Figure 1a–1c). As a consequence, we obtained \mathcal{T}_d ranging from 1.86 (estimated from cumulative cases in N.Y. State) to 2.96 (estimated from cumulative deaths in Switzerland). These

• Endo A. *et al.* (2020) *Wellcome Open Research* 5, DOI: 10.12688/wellcomeopenres.15842.3.
• Lauer S. A. *et al.* (2020) *medRxiv*, DOI: 10.1101/2020.02.02.20020016.

values are much lower than the values reported in early influential studies of Wu *et al.*^(a), Wu *et al.*^(b), and Li *et al.*: 5.2 days, 6.4 days, and 7.4 days, correspondingly. In these studies the basic reproduction number has been estimated to lie in between 1.94 and 2.68.

Response to the Report from Reviewer #3

1 Overall review

The authors present a SEIR model allowing for a more accurate estimation of R_0 with the presence of disease super-spreading in the current SARS-CoV2 pandemic. Particular attention was drawn to the initial lower disease growth spreading phase.

The present SEIR model includes multiple exposed ($m = 6$) and infectious ($n = 2$) stages with a gamma-type Erlang distribution of incubation time and infectious time period, respectively. This approach is known to lead to a more realistic (less optimistic) estimation of the basic reproductive number R_0 based upon the empirical determination of the infection doubling time T_d , when compared to exponential-distribution models, or to models with inadequate assumptions about the disease incubation time period. Stochastic simulations of model dynamics were presented, based upon the use of a small proportion of super- and hyper-spreaders in the general population.

The paper addresses an important aspect of COVID-19 pandemic concerning the need for more robust preventive strategies directed at disease super-spreaders in earlier stages of disease transmission, while focusing on the risks of downplaying those preventive interventions due to underestimation of R_0 .

2 Major points to be addressed

The authors use sound statistical definitions throughout the paper. But several of those definitions lack adequate explanation in the text. The rationale behind the choice of some of the distribution parameters should be presented with enough detail, in order to facilitate the reading task.

Response: In the Results section of the revised manuscript, we briefly introduced the Erlang-type distribution and explained its relation to the ODE representation of the model. In the Methods section, we explained the relation between the latency period, infectious period and generation time, and the relation between generation time and serial interval (under certain assumptions these times are equal).

-
- Wu J. T. *et al.* (2020^a) *Nature Medicine* **26**, DOI: 10.1038/s41591-020-0822-7.
 - Wu J. T. *et al.* (2020^b) *The Lancet* **395**, DOI: 10.1016/S0140-6736(20)30260-9.
 - Li Q. *et al.* (2020) *New England Journal of Medicine*, DOI: 10.1056/NEJMoa2001316.

a) In page 2/12, lines 58–60 Why did the authors consider for analysis the epidemic phase in which the number of registered cases began to exceed one hundred? Why one hundred, and not a different cut-off? A short explanation should suffice. The authors should also address the fact that by comparing countries with different population sizes, that cut-off may induce small differences in proportional disease prevalence rates per 100.000 inhabitants, among different countries.

Response: One hundred for the number of total cases (and, in the revised manuscript, 10 for the number of total fatalities) is a trade-off between two goals: (i) analysis of epidemic progression when stochastic effects associated with individual transmission events, including transmission by super-spreaders, are relatively small (see stochastic simulation trajectories in Figure 2a) and (ii) analysis of the exponential phase of epidemic progression, which is relatively short due to imposition of restrictions (see empirical trajectories in Figure 1a, 1b).

Indeed, there is a problem with comparing large and small countries. Realising this, we decided to consider New York State instead of the USA. Obviously, China is a very populous country, but majority of cases and deaths registered in the exponential growth phase come from a single province (Hubei). In the revised manuscript, we additionally estimated doubling time from cumulative deaths (see new panel B in Figure 1). For cumulative deaths we chose a threshold of 10, for reasons described in the previous paragraph.

b) In page 3/12, line 2 The assumption that super-spreading events are implausible at the beginning of the pandemic is a little tricky. If we look at what locally happened in Bergamo in February/March of 2020, after the game Atalanta–Valencia of the European Champions League, we see that super-spreading could be an isolated single explosive event before any previous spreading. While the epidemic was in its initial phase in Milano (where the game took place) almost 40.000 Atalanta fans went from Bergamo to Milano to watch the game. The super-spreading event that occurred during and after the game was responsible for the later outbreak in Bergamo, when those fans returned home. The population of Bergamo is estimated to be close to 120.000 inhabitants. So, this flash hyper-spreading event precipitated the Bergamo outbreak by instantly infecting close to a third of the town population. In a similar note, the small contingent of 3.000 Valencia fans, was probably also responsible for the earlier disease outbreak in Valencia and later on in Spain, that followed soon after that game. So, super-spreading events may not be that implausible at the initial phase of a pandemic. The authors should address this point presenting evidence of their statement in page 3/12, line 2.

Response: We agree that the game Atalanta–Valencia of the European Champions League, that took place on February 19, 2020, in Milano, was very likely a super-spreading event that accelerated epidemic spread in Lombardy and Spain, and we refer to this example when discussing super-spreading. We may notice, however, that on February 23, 2020, four days after the game, there were 57 cases officially registered in Lombardy. Taking into account that latent/incubation time is about 5 days and that there is delay in reporting, we may expect that these people were infected before the game. As the potential super-spreading events are generally infrequent, there is a little chance that such event is attended by an infectious individual

in the case when the number infectious individuals is very low. The stochastic trajectories shown in Figure 2a indicate that super-spreading events statistically occur when the number of infected individuals is large (the convention is that each trajectory turns red after a first super-spreading event). What is however crucial for our reasoning is the observation that doubling time calculated from the initial phase of the epidemic outbreak (that does not contain super-spreading events) will be unavoidably overestimated.

c) In page 3/12, lines 3–7, the following sentence is unclear:

“We show that in the presence of super-spreaders the median trajectory starting from a single infection grows much slower than the average trajectory, which very likely leads to overestimation of the doubling time.”

The median trajectory of what? The infection rate? The absolute number of cases? The infection growth rate?

Response: Text containing this imprecise sentence has been reformulated.

d) In page 3/12, line 33–34 The meaning of T_d should be defined at this stage, and not later in the same page, in line 44.

Response: Corrected.

e) In page 3/12, line 34–42 The authors should present the rationale for the use of a two and three weeks period for estimation of T_d in the Methods section.

Response: We use two-week periods because this is the longest period for which the growth of the number of cumulative cases as well as cumulative deaths is nearly exponential in all considered countries and N.Y. State. We add this explanation to Methods (subsection (b)). We removed analysis within three-week periods.

f) In page 3/12, line 35 In the present SEIR model, the authors defined the R stage as removed. Although this definition may be partially acceptable, it does not comply with the well accepted definition of stages in the SEIR model in which R stands for recovered and not removed — see Vynnycky E. and White R G. An introduction to infectious disease modelling, Oxford, 2010.

Response: We add the explanation why our fourth stage, ‘R’ or ‘removed’, corresponds to the removed cases, which includes both recovered and deceased individuals, but also currently ill individuals that remain isolated and are thus considered effectively non-infectious (Methods, subsection (a)).

In page 8/12, lines 26–33 The present model is based on a gamma-type Erlang distribution involving the precise determination of the infection doubling time T_d in several world regions affected by the pandemic (China, USA and Europe). This type of distribution is usually considered more realistic than exponential-distribution based SEIR models. Exponentially distributed models reveal an initial slower epidemic growth,

while predicting a smaller peak number of cases and lasting longer than gamma-distribution models. The authors defined the SEIR model with $m = 6$ exposed stages and $n = 2$ infectious stages. These m and n subclasses of exposed and infectious stages, respectively, are usually accepted as efficient computational alternatives for modelling infections with gamma-type probability distributions. However, models with higher values of m tend to reduce the variance of the incubation period, leading to an overestimation of R_0 . In the Method section the authors should present the rationale for particularly choosing $m = 6$ and $n = 2$ in Erlang distributions for the incubation and infectious period, respectively. A simple explanation should be adequate.

Response: Yes, we are aware of this fact and this is why based on available data on incubation time distribution we chose the model with $m = 6$. Having no clear evidence regarding distribution of infectious state duration and keeping in mind that it can depend on protective measures, we considered two model variants with $n = 1$ and $n = 2$ giving somewhat different estimates of doubling time (see Figure 1c). We have added this explanation to the Methods section.

3 Minor aspects

h) In page 4/12, lines 41-42 The following sentence is not precise and should be rephrased.

“Our calculations suggest the range of R_0 that is higher than initially estimated (Boldog et al., 2020; Liu et al., 2020).”

Our suggestion:

“Our calculations suggest the range of R_0 to be higher than initially estimated (Boldog et al., 2020; Liu et al., 2020).”

Response: This sentence has been rephrased.

i) In page 7/12, line 30 In the following sentence, we suggest replacing the word expect by a more adequate expression (eventually, the word suspect could be used instead; other sound alternatives may also be acceptable)

“Consequently, we expect that...”

Response: Corrected.

Conclusion

The present paper is scientifically sound, focusing on a relevant aspect of COVID-19 early transmission that should be critical in defining better preventive strategies by public health institutions at country and regional level.

Response: Thank you for your helpful comments and favourable assessment.

□

Appendix C

Royal Society Open Science review

Manuscript ID:
RSOS-200786.R1 (revised by the authors)

Title:
Super-spreading events initiated the exponential growth phase of COVID-19 with R_0 higher than initially estimated

1 Revised version overview

The authors made substantial and adequate improvements.

We briefly highlight important points addressed in the revision process:

1.1. Elimination of the analysis based upon the three-week interval exponential phase - simplification of data resulting in more homogeneous results.

1.2. The use of New York state data in substitution of overall USA heterogeneous results. It is an adequate improvement.

1.3. Estimation of T_d from the increase in cumulative cases and cumulative deaths. The under-reporting of deaths should, in theory, be less significant.

1.4. Figure 1 is clearer.

1.5 The inclusion of generation and serial intervals analysis is an adequate improvement.

1.6. Proper explanations were given, with the adequate detail, in the Methods and Results sections, concerning our initial doubts and requests.

2 Minor suggestions

a)

In page 14/38, line 40

The following sentence lacks a *comma*.

super-spreading events occur and fast...

Our suggestion:

*super-spreading events **occur, and** fast...*

b)

In page 14/38, line 42

The verbal conjugation lacks concordance.

cases or deaths has been exceeded...

Our suggestion:

*cases or deaths **had** been exceeded...*

c)

In page 15/38, line 48

The verbal expression could be improved.

...that overlaps with...

Our suggestion:

*...**overlapping** with...*

d)

In page 17/38, line 29 - Table 1

Word misspelling.

...mean searial period...

Our suggestion:

*...mean **serial** period...*

e)

In page 17/38, line 57

The verbal conjugation could be improved.

...registered cases has been evident...

Our suggestion:

*...registered cases **had** been evident...*

f)

In page 18/38, line 33

The verbal conjugation could be improved.

...below 1) has not been yet...

Our suggestion:

*...below 1) **had** not been yet...*

g)

In page 19/38, line 42

The verbal conjugation could be improved.

...figure 3 we provided extinction probability...

Our suggestion:

*...figure 3 we **provide** extinction probability...*

h)

In page 19/38, line 46

Missng a connecting particle.

...be equal 3.05 days...

Our suggestion:

*....be equal **to** 3.05 days...*

3 Conclusion

The revised version is clearer. Major issues of concern were adequately attended by the authors.

Tomasz Lipniacki
Department of Biosystems and Soft Matter
Institute of Fundamental Technological Research
Polish Academy of Sciences

Pawińskiego 5B, 02-106 Warsaw, Poland
Phone: +48 22 8261280
E-mail: tlipnia@ippt.pan.pl
Website: <http://pmbm.ippt.pan.pl>

Subject: **Final revision of manuscript RSOS-200786.R1**

August 26, 2020

Dear Reviewers, Dear Editor,

We are pleased to learn that our revised manuscript “Super-spreading events initiated the exponential growth phase of COVID-19 with R_0 higher than initially estimated” has been provisionally accepted for publication in *Royal Society Open Science* subject to minor revision. In the revision we have made to meet the final acceptance requirements, we followed all the suggestions of Referee 3, as deemed necessary. We also responded to further comments of Referee 1, which resulted in inclusion of a new table in the main text (Table 2) and amending the electronic supplementary material with a new figure (Figure S2). Our responses to specific comments are given below.

Sincerely,

Tomasz Lipniacki

Professor and Head

Response to the Report from Reviewer #1

I appreciate the authors' effort to improve the readability of the paper and also clarification regarding my previous comments. However, I still have a few concerns which I suggest the authors consider. I think these are important to render the paper useful and interpretable.

As I understand, the main messages of this paper are the following:

- (i) The basic reproduction number of SARS-CoV-2 is estimated to be 4–11.*
- (ii) This is higher than existing estimates because of 6 exposed states and doubling time used was shorter than previously used.*
- (iii) A possible reason for the shorter estimates of doubling time is superspreading events absent in the earlier phase; simulation shows a change in epidemic growth after the emergence of the first super/hyper spreader.*
- (iv) The authors' approach using 14-day-case counts from the day the cumulative number of cases exceeds 100 is more reliable in estimating R_0 in the presence of superspreading.*

Response: Thank you for the positive assessment of our revision. Yes, the above four points accurately summarise our main findings.

Below I detail my comments for each point.

(i) Now we know that R_0 of SARS-CoV-2 substantially varies between countries, and also even between studies on the same country depending on data and methods used (see Rahman et al. Rev Med Virol, for example). Some already reported R_0 estimates of 4–6 in some countries (although whether these are reliable remains an open question). Given that the authors apparently picked up countries with fastest epidemic growth (many other countries had a slower initial growth), I wonder if treating the value of 4–11 to be R_0 in a general context without specifying the country is a good idea.

Response: The Referee is right in saying that we have picked countries/regions with fast epidemic growth, as we expect that in other countries the slower growth was a consequence of protective measures, in particular banning mass gatherings. Nevertheless, we revised the abstract to make it more precise. The key statement currently reads: “Here we used a SEIR model that properly accounts for the distribution of the latent period and, based on empirical estimates of the doubling time in the near exponential phases of epidemic progression in China, Italy, Spain, France, United Kingdom, Germany, Switzerland, and New York State, we estimated that R_0 lies in the range 4.7–11.4.”

(ii) It is nice to have Table R1, but these studies are looking at Wuhan/Hubei with only a limited number of initial cases (therefore inevitably suffer small sample size). Country/region names should be specified to avoid misunderstanding, and also these T_d values should be only compared against estimates for the same country/region. Even if there was no such existing study for a country, I think the authors could use JHU data before the periods they analyzed in this paper. Indeed, it would be very interesting to see if there was a change in growth rates in the early phase of an epidemic as suggested in Fig 2, which will add empirical evidence to the authors’ argument.

Response: Yes, all T_d estimates given in Table 1 are based on epidemic development in Hubei; we add this information to the table caption. We have followed the interesting suggestion to estimate T_d from the early data in eight considered regions — see Table 2 and text at the end of the Results section. To be in line with the analysis in Figure 3, we compared estimates using ‘30 days since the 1st case’ method with the estimates of ‘14 days since 100 cases’ method (the latter is the primary method for our study). As expected, T_d estimates using ‘30 days since the 1st case’ method in most cases are larger and more dispersed (in range 1.92–12.6) than the estimates based on the primary method (1.86–2.88). As suggested by the Reviewer, we used JHU data, except for China for which JHU does not provide data for the first month of epidemic. Also, because of missing data for three other countries, we started counting from the second, not the first, case.

(iii) – I concur with Reviewer 3 pointing out that $m = 6$ and $n = 2$ are arbitrary and should be justified. For example, gamma distribution was used to estimate incubation period in some study, and could the authors confirm that the shape of distributions in these existing studies are similar to the one used here, or at least the mean and standard deviation are in line?

Response: The Reviewer may notice that we use two model variants with $n = 1$ and $n = 2$ which gave us the lower and upper estimate of the basic reproduction number (Fig. 1). To the best of our knowledge, in the context of COVID-19, other values of n have not been used. The Erlang distribution of latent period with average 5.28 days and shape coefficient $m = 6$ is in line with published epidemiological data, as shown in new electronic supplementary figure S2.

– Emergence of hyperspreader: As I mentioned earlier, one way to add support to this hypothesis is to compare this with actual data in the initial phase. Also as in my initial comment, 1% of hyper spreaders who is 200 times infectious than average in Fig 2 is quite extreme and I am not sure if this phenomenon could happen in reality. In the response the authors agreed that the data on superspreading is still limited and thus a range of values needs to be tested. How about more realistic scenarios, such as moderate proportion and infectiousness of superspreaders and/or non-binary heterogeneity represented by a negative binomial distribution with small overdispersion parameter, which now is empirically observed worldwide (for example, Ramanan et al., Sun et al. both in medRxiv)?

Response: The case with 1% of hyperspreaders being 200 times more infectious than normal spreaders is considered as an extreme case in Fig. 3. We agree that non-binary heterogeneity is much more realistic, but it is also harder to parametrise. As the ban on mass gathering was the first measure implemented in most countries, it is hard to expect that data gathered on superspreading is accurate. The large dispersion of T_d values estimated using ‘30 days since the 1st case’ method (new Table 2) suggests that in some regions huge superspreadings took place while in others did not.

(iv) I agree that data in the later period is more reliable to estimate R_0 (especially when there is superspreading). However, the comparison here seems unfair to me because the sample size and time needed to produce an estimate is different between the two approaches. Using initial data is more prone to bias and estimation error, but it may have to be accepted as a necessary trade-off in the earliest analysis. If we have to wait 14 days after the first 100 cases, with a doubling time of 2, the total number of cases could go up to $> 10,000$ which is too late to take action for epidemic control (and if intervention comes in place the estimated reproduction number is out of date). For a fair comparison, I suggest the authors show the difference in time of estimation and the number of cases accumulated during this time gap and discuss the strengths and weaknesses of each approach. Also note that information on the degree of superspreading is extremely difficult in the initial phase when R_0 estimates are most needed.

Response: We are grateful for this comment as we have not thought about our results and other existing results from the perspective of the trade-off between accuracy of the reproduction number estimate and the

time at which such estimate was possible. It is natural that in later time, when more data are available, more accurate estimates are possible. In Table 2, where we compare estimates made with the ‘30 days since the 1st case’ method and with the ‘14 days since 100 cases’ method, we also give the day of the year in which a given estimate became possible, as well as the number of cumulative registered cases on that day. This shows that, indeed, the second method (in most cases) requires longer time span. It is however important to notice that for China the ‘14 days since 100 cases’ method estimate (chosen in our study and giving reproduction number in range 5.6–7.3) was possible on February 4, 2020, before the surge of the epidemic in Europe and US, and more than one month ahead of the first European country-wide lockdown imposed in Italy (March 9, 2020). From a broader perspective our analysis suggests that the reproduction number can be severely underestimated based on the early epidemic data.

Minor comment

– Fig 1: Month names should be in English

Response: This has been corrected, thank you.

Response to the Report from Reviewer #2

Authors have made the changes based on the reviewers comments. I have no further comments.

Response: Thank you for your favourable assessment.

Response to the Report from Reviewer #3

1 Revised version overview

The authors made substantial and adequate improvements.

We briefly highlight important points adressed in the revision process:

- 1. Elimination of the analysis based upon the three-week interval exponential phase – simplification of data resulting in more homogeneous results.*
- 2. The use of New York state data in substitution of overall USA heterogeneous results. It is an adequate improvement.*

3. *Estimation of T_d from the increase in cumulative cases and cumulative deaths. The under-reporting of deaths should, in theory, be less significant.*
4. *Figure 1 is clearer.*
5. *The inclusion of generation and serial intervals analysis is an adequate improvement.*
6. *Proper explanations were given, with the adequate detail, in the Methods and Results sections, concerning our initial doubts and requests.*
7. *Minor suggestions were included in the attached file.*

[SUGGESTIONS FROM THE ATTACHED FILE]

2 Minor suggestions

a) In page 14/38, line 40 The following sentence lacks a comma.

super-spreading events occur and fast...

Our suggestion: super-spreading events occur, and fast...

b) In page 14/38, line 42 The verbal conjugation lacks concordance.

cases or deaths has been exceeded...

Our suggestion: cases or deaths had been exceeded...

c) In page 15/38, line 48 The verbal expression could be improved.

...that overlaps with...

Our suggestion: ...overlapping with..

d) In page 17/38, line 29 Table 1 Word misspelling.

...mean searial period...

Our suggestion: ...mean serial period...

e) In page 17/38, line 57 The verbal conjugation could be improved.

...registered cases has been evident...

Our suggestion:...registered cases had been evident...

f) In page 18/38, line 33 The verbal conjugation could be improved.

...below 1) has not been yet...

Our suggestion: ...below 1) had not been yet...

g) In page 19/38, line 42 The verbal conjugation could be improved.

...gure 3 we provided extinction probability...

Our suggestion: ...gure 3 we provide extinction probability...

h) In page 19/38, line 46 Missng a connecting particle.

...be equal 3.05 days...

Our suggestion:be equal to 3.05 days...

Response: We are grateful for your careful reading. Manuscript has been corrected in all suggested places.

3 Conclusion

The revised version is clearer. Major issues of concern were adequately attended by the authors.

Response: Thank you for your favourable assessment.